# The cerebellum contributes to generalized seizures by altering activity in the ventral posteromedial nucleus

Jaclyn Beckinghausen [1,2,3], Joshua Ortiz-Guzman[3,4], Tao Lin [1,3], Benjamin Bachman[5], Luis E. Salazar Leon[1,2,3], Yu Liu [6], Detlef H. Heck[6], Benjamin R. Arenkiel [2,3,4,5] & Roy V. Sillitoe [1,2,3,4 ✉]

Thalamo-cortical networks are central to seizures, yet it is unclear how these circuits initiate seizures. We test whether a facial region of the thalamus, the ventral posteromedial nucleus (VPM), is a source of generalized, convulsive motor seizures and if convergent VPM input drives the behavior. To address this question, we devise an in vivo optogenetic mouse model to elicit convulsive motor seizures by driving these inputs and perform single-unit recordings during awake, convulsive seizures to define the local activity of thalamic neurons before, during, and after seizure onset. We find dynamic activity with biphasic properties, raising the possibility that heterogenous activity promotes seizures. Virus tracing identifies cerebellar and cerebral cortical afferents as robust contributors to the seizures. Of these inputs, only microinfusion of lidocaine into the cerebellar nuclei blocks seizure initiation. Our data reveal the VPM as a source of generalized convulsive seizures, with cerebellar input providing critical signals.

[1] Department of Pathology and Immunology, Baylor College of Medicine, Houston, TX, USA. [2] Department of Neuroscience, Baylor College of Medicine, Houston, TX, USA. [3] Jan and Dan Duncan Neurological Research Institute of Texas Children's Hospital, 1250 Moursund Street, Suite 1325 Houston, TX, USA. [4] Program in Developmental Biology, Baylor College of Medicine, Houston, TX, USA. [5] Department of Molecular and Human Genetics, Baylor College of Medicine, Houston, TX, USA. [6] Department of Biomedical Sciences, University of Minnesota Medical School, Duluth, 103515 University Dr., Duluth, MN, USA. ✉email: sillitoe@bcm.edu

At their worst, recurrent seizures can be fatal, and at best, they significantly decrease a patient's quality of life. While some individuals experience only one or a few isolated seizures that may not be classified as a standalone condition, those that experience chronic or recurrent unpredictable seizures are diagnosed with epilepsy[1]. About a third of patients with epilepsy experience persistent seizures even after treatment[2], and in over 60% of cases, the catalyst for initial seizure precipitation is unclear[3]. As a consequence, it is difficult to optimize therapeutic options. Thus, there is a pressing need to define the relationship between abnormal neural activity and seizure-induced behavior, whether the seizure exists as an isolated event or as part of a chronic epileptic condition.

Epilepsy encompasses of a variety of seizure presentations and classifications, ranging from rigidity and behavioral arrest to dramatic, rapid motor convulsions. According to the International League of American Epilepsies (ILAE), seizures are first categorized by onset (focal onset, generalized onset, or unknown onset), followed by type (focal, generalized, combined focal and generalized, or unknown). Finally, a diagnosis of epilepsy syndrome may be made if the patient exhibits a co-occurrence of overlapping seizure-type features[1]. Of the more than 25 categories of seizures within epilepsy, those with motor convulsions are perhaps the most disruptive and generally feared among patients and caregivers. Unfortunately, these are also the most common class of generalized seizures[4].

The reciprocal cortico-thalamic loop has been widely studied for its unequivocal role in seizures: however, the precise role of individual thalamic nuclei and how their unique combination of inputs promotes the generation of abnormal brain activity is still poorly understood. Lesioning, pharmacological inactivation, and optogenetics studies of thalamic somatosensory regions and their cortical targets demonstrate a requirement for these areas in both initiation and maintenance of certain epilepsy phenotypes[5–10]. Furthermore, although currently understudied, rodent studies specifically point to brain networks carrying facial information as heavily involved in seizures. The possibility of an epileptic origin in peri-oral circuits is especially intriguing for generalized motor seizures, as facial clonus is one of the first behavioral readouts observed in this dynamic phenotype. We therefore chose to investigate the origin of generalized, convulsive seizures9 by testing circuits in thalamic regions that are known to be tightly interconnected with the facial somatosensory cortex.

The ventral posteromedial nucleus of the thalamus (VPM), a region suspected to be involved in absence and focal seizures[11,12], is a compelling candidate for initiating generalized motor seizures. Directly connected to regions of the somatosensory cortex that are necessary and sufficient for absence ictal activities[8], the VPM may be effective in aborting post-stroke epileptic seizures[13]. Importantly, there is often a continued progression from absence seizures to clonic and eventually tonic-clonic seizures. Thus, brain regions known to be essential in absence seizure generation or maintenance (such as the VPM) may be directly relevant to generalized seizures with motor convulsions. Using optogenetics, we tested whether activation of inputs specifically to the VPM, but not to other surrounding thalamic nuclei, drives severe tonic-clonic seizures. We then used virus tracing to identify the specific neural projections that may contribute to ictogenesis: we were specifically interested in examining neural projections that are monosynaptically connected to the VPM.

We identify the cerebellum as one such source of direct projections to the VPM. Although the VPM has previously been implicated in seizure pathophysiology, its definitive role in seizures has been widely debated[14–17]. Based on historical evidence suggesting that the cerebellum provides an inhibitory influence in seizures, Irving S. Cooper used the cerebellar cortex as the very first deep brain stimulation target in epilepsy treatment. Despite multiple cases of >50% seizure reduction in these patients, variability in subsequent trials led to a shift in attention away from the cerebellum for therapeutic intervention in this condition[16,17]. Still, cerebellar pathophysiology is consistently observed in patients with numerous disorders—including Unverrict Lundborg Disease and familial cortical myoclonic tremor with epilepsy—containing a common denominator: seizures[18–20]. Here, we investigated the role of the VPM in seizure initiation and propose that cerebellar input to this region is essential for its generation. We investigated the mechanisms of seizure initiation through the VPM by asking (1) how thalamic neurons behave during generalized seizure activity, and (2) whether the cerebellum contributes substantially to the initiation of generalized seizures in our optogenetic mouse model.

## Results

### Optogenetic stimulation of the VPM elicits tonic-clonic seizures in $Ntsr1^{Cre}$;$ChR2$-$EYFP$ mice.

The thalamus is a well-known player in the seizure network, although the contribution of its different subregions to mediating seizures is not well defined. To optogenetically target neurons known to be involved in the seizure network—i.e., the thalamus and cortex—we used a previously established $Ntsr1^{Cre}$ transgenic mouse line[21]. $Ntsr1$ encodes for a protein that is expressed primarily in the gastrointestinal tract, but expression is also reported across different brain regions including layers 5 and 6 of the cerebral cortex, the thalamus, ventral tegmental area, and the cerebellar nuclei[22–25]. To confirm and further characterize its expression profile, we crossed $Ntsr1^{Cre}$ male mice to two different reporter lines: one that expresses Sun1 ($Ntsr1^{Cre}$;$ROSA^{lox-stop-lox-Sun1}$), a small nuclear envelope protein (Fig. 1a–e), and the other that expresses TdTomato ($Ntsr1^{Cre}$;$ROSA^{lox-stop-lox-TdTomato}$), which fully illuminates cellular architecture including distal axons located in downstream brain regions (Fig. 1f–j). In regards to the thalamic expression, we observed that the fluorescent emission from the thalamus, though robust in the TdTomato line (Fig. 1g), was missing in our Sun1 reporter line (Fig. 1b). Staining with calbindin (Supplementary Fig. 1a–i), a calcium binding protein that labels unique populations of neurons in different regions of the brain, and NeuN, a pan-neuronal marker (Supplementary Fig. 1j–o), demonstrated no colocalization with Cre-driven TdTomato: instead, the genetic reporter is evident in the peri-neuronal space (Supplementary Fig. 1h, n). The structure of the labeled profiles coupled with the absence of Sun1 thalamic reporter expression in the entire thalamus indicated that the fluorescence observed in the thalamus of the TdTomato mice likely reflects incoming axons and fibers of passage that originate in other areas where $Ntsr1^{Cre}$ is expressed, such as the cerebellar nuclei (Fig. 1j).

We also observed that $Ntsr1^{Cre}$ expression within the cerebellar nuclei was not evenly distributed, as the medial cerebellar nuclei (fastigial nucleus) contained very few reporter-expressing neurons (Fig. 1d). Instead, we found that the reporter signal was heavily localized to projection neurons of the interposed (central) and dentate (lateral) nuclei. Of 6 coronal sections from 3 animals, we identified 84.89% (545 of 642 neurons) recombination efficiency (calculated as the number of Cre-induced reporter positive neurons divided by the number of NFH-positive neurons) in the interposed and dentate nuclei versus only 9.6% (39 of 407 neurons) in the fastigial nuclei. This is an important consideration when manipulating cerebellar output, as each nucleus has distinct downstream targets[26–28]. Between each nucleus, we did not notice distinct morphological differences in $Ntsr1^{Cre}$ positive neurons, although additional characterization

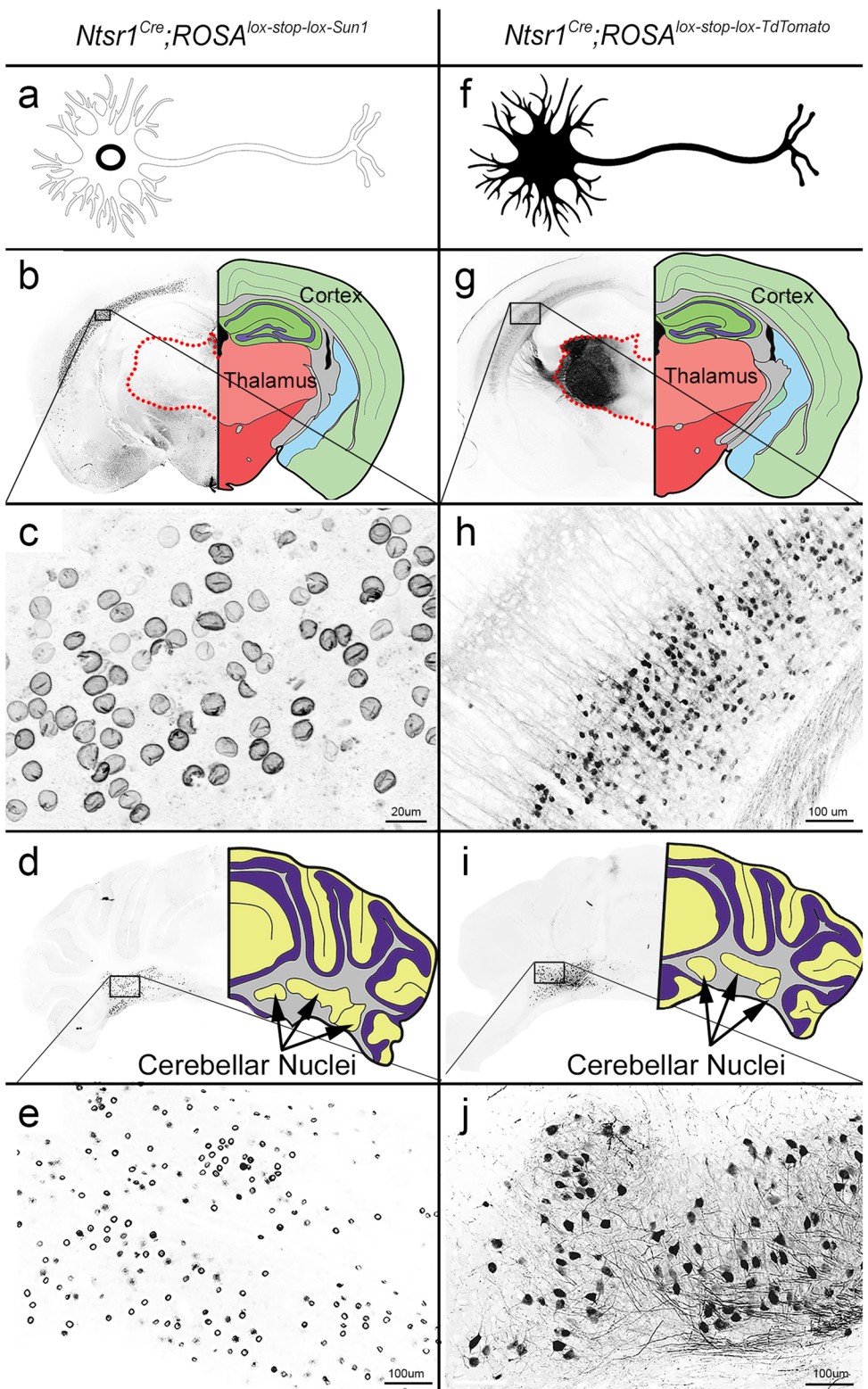

using molecular markers would be required to parse out yet unappreciated differences in these neurons.

To optogenetically manipulate these genetically marked neurons, we next crossed the *Ntsr1*$^{Cre}$ mice to the *ROSA*$^{lsl-Channelrhodopsin-EYFP}$ (hereafter referred to simply as ChR2-EYFP) reporter strain. The offspring of this cross carried light-sensitive neurons in the aforementioned *Cre*-expressing brain regions as well as in the axons projecting to, or passing through, the thalamic nuclei. We delivered

465 nm light to the thalamus via surgically implanted fiber optics to indirectly modulate thalamic neurons by activating the axons and terminals of their presynaptic, ChR2-expressing afferent inputs (including the thalamic, cerebellar nuclei, cortical, and striatal fibers, Supplementary Fig. 2a–e, 2a, b). We delivered pulses of 495 nm LED light to the thalamic ventral posteromedial nucleus (VPM, Fig. 2b) for a total of 1 h per day for one week, toggling between 30 s light-on and 60 s light-off. Of multiple frequencies tested (1–130 Hz), we

**Fig. 1 *Ntsr1^Cre* is expressed in corticothalamic layer 6, cerebellar nuclei, and the axons and fibers of passage within the thalamic nuclei. a** Schematic of Sun1 expression, which labels the nuclear envelope of Cre-expressing neurons. **b** Expression of Sun1 is seen in layer 6 of the cerebral cortex (black box, high power image in (**c**)), but not in thalamic nuclei (red dotted outline). **c** High power image of cortical neurons expressing nuclear Sun1. Scale bar = 20 μm. **d** Cerebellar nuclei neurons in the dentate and interposed nuclei express Sun1. **e** High-power image of Sun1 in the cerebellar nuclei, showing the specificity of fluorescence for the nuclear membrane. Scale bar = 100 μm. **f** Schematic of TdTomato expression, which fills the entire cell including dendrites and axonal processes. **g** Expression of TdTomato is observed in layer 6 of the cerebral cortex (black box, high power image in (**h**)) as well as in the thalamus (red dotted outline). **h** High power image demonstrating TdTomato expression in cortical layer 5/6 neurons. Scale bar = 100 μm. **i** Cerebellar nuclei neurons in the dentate and interposed nuclei express TdTomato. **j** High power magnification of cerebellar nuclei neurons with TdTomato fluorescence in the somas as well as in axons exiting the cerebellum. Scale bar = 100 μm. LUT (color look up table) has been inverted in all fluorescent images (**a–j**). For panels (**b**), (**d**), (**g**), and (**i**), anatomical drawings were recreated in photoshop based upon the Allen Brain Atlas, mouse.brain-map.org.

found that daily unilateral delivery of 30 Hz pulses of LED light at ~3.6 mW to the VPM consistently elicited robust, generalized seizures ($n = 24$ biologically independent mice, Supplementary Video 1) involving forelimb and facial clonus, squeaking/vocalizations, body convulsions, and in some cases, drooling and eventual loss of posture. When a seizure was successfully induced, it outlasted the 30 s light delivery, indicating that the seizure itself becomes independent of the optogenetic paradigm once established.

While seizures were reliably evoked when fibers were precisely targeted to the VPM, light delivery to the surrounding nuclei, including the posterior thalamic nucleus (PO, $n = 3$ biologically independent mice), the ventroposterior lateral nucleus (VPL, $n = 3$ biologically independent mice), the reticular thalamic nucleus (RT, $n = 3$ biologically independent mice), and the lateral dorsal nucleus (LD, $n = 2$ biologically independent mice) did not elicit overt behavioral changes (example of the VPL stimulation is shown in Supplementary Video 2) (Fig. 2c). We quantified the relative fluorescence in each of these individual thalamic nuclei ($n = 5$ biologically independent mice, 5–10 slices per mouse) to determine whether the non-responsive nuclei received less ChR2-expressing input. Of note, our *TdTomato* and *ChR2-EYFP* lines exhibited a similar pattern of fluorescence within thalamic nuclei: that is, uneven distribution of fluorophore intensity. This is not readily evident in our TdTomato expression figure (Fig. 1g), as oversaturation of thalamic nuclei was necessary to visualize brain-wide regions with low, but still positive Cre-induced reporter expression. While the fluorophore intensity was useful in our TdTomato line for characterization, we chose to quantify the ChR2-EYFP signal from our optogenetic line since these were our subjects for seizure-inducing experiments. Some nuclei, such as the RT, PO, and CL indeed had significantly less fluorescence (Dunnett's multiple comparisons test, Fig. 2a, d), which may account for the absence of behavior upon their stimulation. However, other nuclei like the VPL contained comparable levels of ChR2-EYFP input as the VPM, yet also failed to recapitulate seizure behaviors upon their stimulation (Supplementary Video 2, Fig. 2c, d). These data suggest that the initiation of seizures in *Ntsr1^Cre;ChR2-EYFP* mice requires the modulation of a critical composite of inputs that converge within the VPM (and therefore the placement of the fiber impacts a large number of responsive VPM neurons) rather than the activation of any combination of circuits within the thalamus, or specific subsets that are less important to initiation of the seizure behavior.

We next compared the phenotype of our optogenetics model with the well-established, chemo-convulsant kainic acid (KA) model ($n = 4$ biologically independent mice). KA is a nondegradable glutamate analog that, when injected intracranially or systemically in vivo, induces behavioral seizures and neuronal excitotoxicity[29]. Although KA-induced seizures are understood to reflect temporal lobe epilepsy, a specific but extremely common form of epilepsy primarily involving cell death of

specific neurons within the hippocampus[30], the behavioral signature is nonetheless reflective of generalized convulsive seizures. We therefore used this model strictly as a basis for phenotypic comparison of a rodent seizure rather than as a parallel for mechanisms underlying the seizure initiation. Indeed, comparison of optogenetic VPM activation and KA injection revealed a number of behaviors common to both models during severe seizure episodes (Supplementary Video 3, Fig. 2e). Specifically, both models exhibited tensing of the body (stage 1 according to a modified Racine scale[31], a standardized method of rating rodent seizure severity—see Table 1), hunching posture, erect Straub tail, facial and forelimb clonus (stages 2–4), and eventual loss of upright postures (stage 5) with convulsions leading to erratic jumping (stage 6). These mirrored behaviors provided evidence of a comparative basis for validating our phenotype as one of generalized seizures: the question we next addressed was how these behaviors relate to brain activity.

**Electrocorticography (EcoG/EEG) recordings reflect a widespread presence of seizure activity in optogenetic seizures.** Electroencephalography (EEG) is an electrophysiological measure used for diagnosing and distinguishing seizure conditions in humans and rodent models. This technique measures population-wide neuronal activity using a series of electrodes placed on the surface of the scalp. In mice, increased signal quality is achieved by placing the electrodes underneath the bony skull and directly above the brain surface. Due to this difference in electrode position, this specific technique is referred to as electrocorticography (EcoG).

To confirm the presence of seizures in the *Ntsr1^Cre;ChR2-EYFP* mice, we recorded EcoG signals from the ipsilateral somatosensory cortex, bilateral motor cortex, and the cerebellum (Fig. 3a). In our analyses, ipsilateral measurements refer to the electrode location relative to the side of VPM stimulation. We tested whether VPM activation in the *Ntsr1^Cre;ChR2-EYFP* mice results in neuronal hypersynchronization, a diagnostic feature of seizures. In accordance with the presentation of seizure behaviors, EcoG activity in *Ntsr1^Cre;ChR2-EYFP* mice with 30 Hz optogenetic stimulation to the VPM exhibited high amplitude, hypersynchronized activity across the cerebral cortices and cerebellum (Fig. 3b, $n = 11$ biologically independent mice). We observed similar EcoG responses in the KA model (Fig. 3c, $n = 4$ biologically independent mice). This hallmark seizure activity was consistent across all optogenetically-induced mice that exhibited seizures with a visible behavioral readout. In Supplementary Fig. 3, we detailed the progression of behaviors seen in these mice in relation to its corresponding ipsilateral cortical EcoG trace (Supplementary Fig. 3). The presence of high amplitude and hypersynchronized EcoG activity and the resemblance of the induced behaviors to an established model, the KA model,

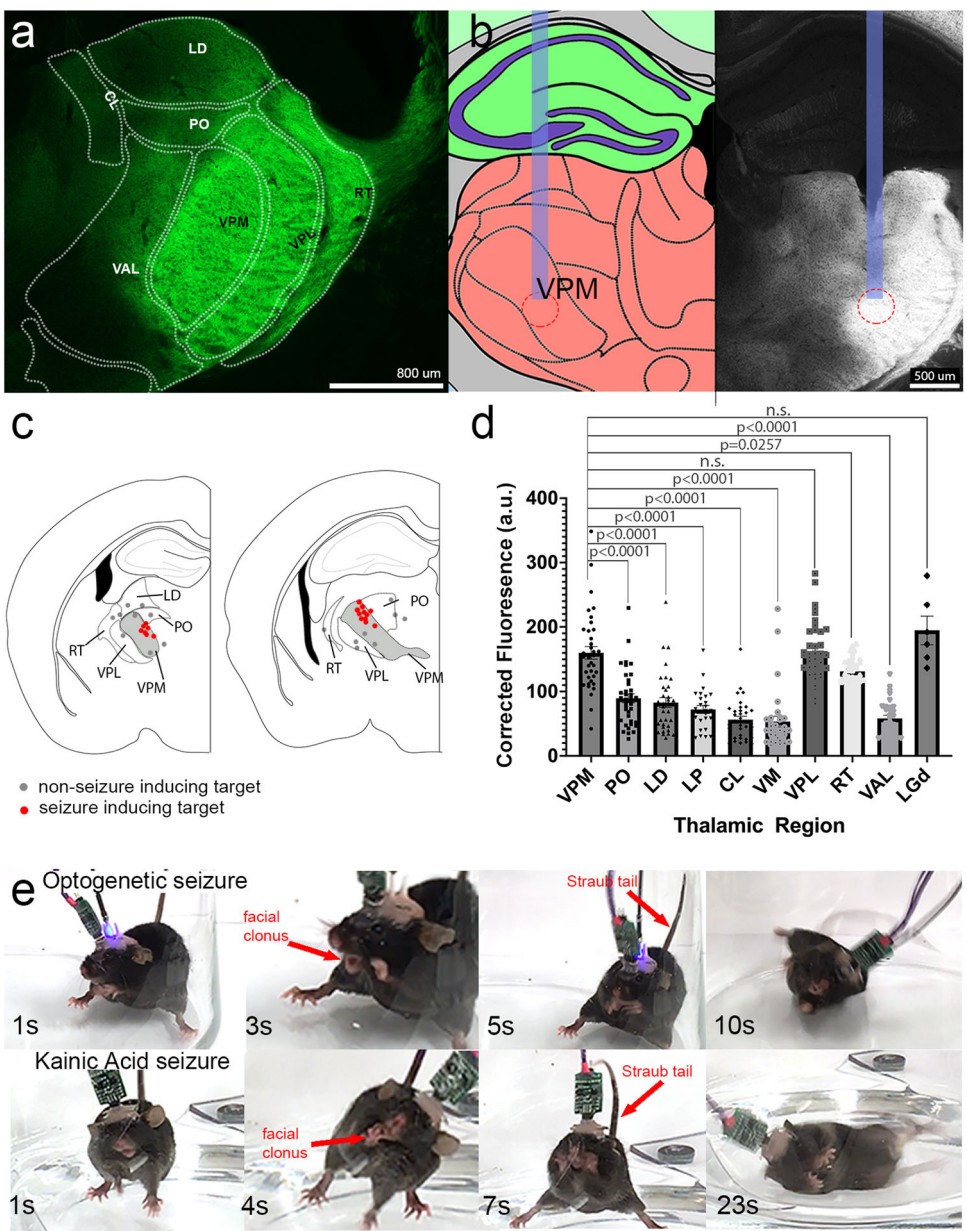

**Fig. 2 Seizure-inducing stimulations are specific to the VPM and behaviorally mimic KA seizures. a** Representative segmentation of thalamic nuclei (coronal) showing relative fluorescence of various nuclei in *Ntsr1^Cre;ROSA^lox-stop-lox-ChR2-EYFP* mice. **b** Example of the optic fiber targeting to the VPM as determined by electrode tracks and photobleaching of the tissue. Anatomical drawing was recreated in photoshop based upon the Allen Brain Atlas, mouse.brain-map.org. **c** Targeting of the fiber optics are examined in mice at the end of experimentation. Each red dot represents the end of a fiber in a single mouse that successfully exhibited stage 6 seizures. Gray dots depict targeting that did not elicit any behavioral change (no seizures). 6 of 24 seizure-induced mice are not shown, as their electrode tracks could not be confidently distinguished post hoc. Anatomical drawing was recreated in photoshop based upon the Allen Brain Atlas, mouse.brain-map.org. **d** Quantification of fluorescent intensity of various thalamic nuclei, some of which are segmented in (**a**), demonstrates that the specificity of VPM as a seizure locus is not due to biased Cre expression, as no behavior is elicited even when VPL or RT is targeted. A Dunnett's multiple comparison's test determined significance between various thalamic nuclei compared to the VPM, df = 38.04 (*p* values for Dunnett's test are shown in the figure). Error bars represent SEM. **e** Still time-lapse images of the seizure phenotype in the optogenetic (above) and KA (below) models of seizure induction demonstrate parallels in behavior beginning with repetitive facial and forelimb clonus and ending in a loss of posture during convulsions. Red arrows point to examples of clonus and an erect, Straub tail, both defining features of rodent seizures. Some images are enlarged to show the phenotypes.

provided compelling evidence that our optogenetic paradigm indeed induces generalized motor seizures.

**30 Hz optogenetic activation of the VPM leads to coherence of cerebral cortical activity.** The synchrony exhibited during seizure-induced EcoG activity reflects the summed dipole fields of the contributing neurons in the local electrode region[32]. While it has been argued that this synchrony is driven by only a sub-population of synchronously co-active neurons[21], it is nonetheless a consistent measure of seizure activity. This activity is typically highly rhythmic, covering a broad spectrum of frequencies[33]. To evaluate the synchronization of rhythmic seizure activity between brain regions in our model, we examined

**Table 1 Modified Racine Scale is used to categorize and rank seizure behavior.**

**Modified Racine Scale**

| Scale | Behavior/Intensity |
|---|---|
| 0 | No change |
| 1 | Sudden behavioral arrest and/or motionless staring |
| 2 | Facial jerking with muzzle or muzzle and eye |
| 3 | Neck jerks |
| 4 | Clonic seizure in a sitting position |
| 5 | Convulsions including clonic and/or tonic-clonic seizures while lying on the belly and/or pure tonic seizures |
| 6 | Convulsions including clonic and/or tonic-clonic seizures while lying on the side and/or wild jumping |

Using a modified Racine Scale adapted from Lüttjohann et al.[31], we compared the severity of seizure phenotypes by ranking different categorical behavioral attributes of the mice upon seizure induction with optogenetic stimulation.

the coherence of 0–50 Hz EcoG oscillations between the motor cortices, ipsilateral somatosensory cortex, and the cerebellum (Fig. 4a–e, Table 2, $n = 3$ biologically independent mice) after seizure initiation. While this method of analysis cannot resolve the precise order of regional activation, it does provide insight as to which areas exhibit an increase in synchrony during seizures.

The ipsilateral motor cortex and ipsilateral somatosensory cortices, both having direct neuronal connectivity with, and physically closest to, the stimulation site (ipsilateral VPM), were most reflective of the intense activity at the seizure origin and exhibited the highest coherence with one aNother both before (black), and during (blue) seizures (Fig. 4b). The contralateral motor cortex also increased in coherence with its contralateral counterpart, though to a lesser degree (Fig. 4c). However, even this lower degree of increased synchronization indicates the presence of a generalized spread of the seizures away from the origin, which ultimately incorporates contralateral brain regions

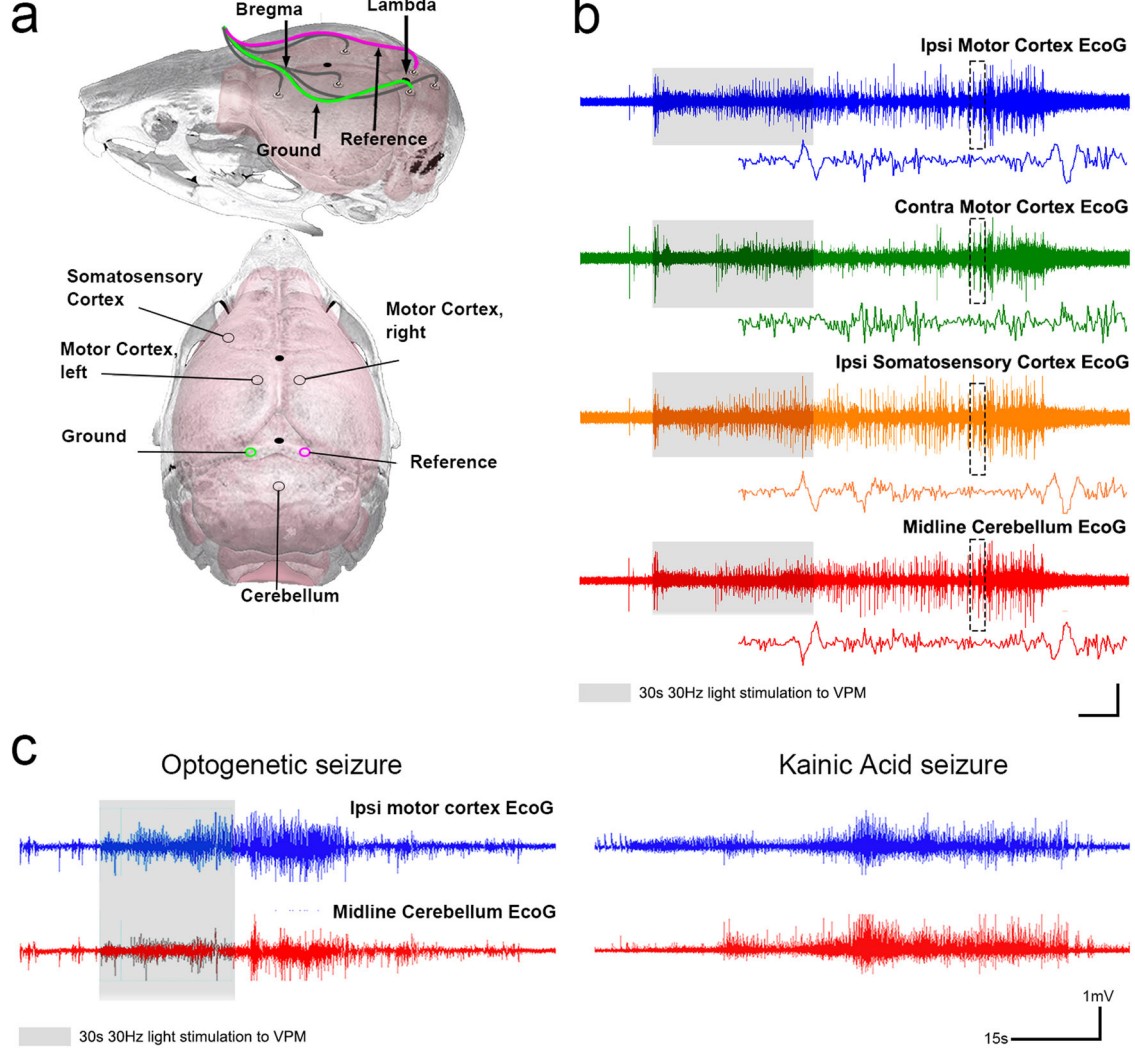

**Fig. 3 EcoG activity in *Ntsr1^Cre^;ChR2-EFYP* mice confirms seizure behavior. a** Schematic of electrode placement for EcoG recordings based on stereotaxic coordinates from the skull landmarks bregma and lambda. Brain overlay adapted from Allen Brain Atlas via Brain Explorer 2. **b** Example of EcoG signals from all four recorded brain regions during unilateral VPM (30 Hz) stimulation shows high amplitude, synchronous activity during the optogenetic stimulation period (gray shaded area). Dashed box captures a region that is expanded below each respective trace to show EcoG activity. Horizontal scale bar is 8 s for the longer traces, 0.5 s for the zoomed insets. Vertical scale bar is 0.5 mV for the longer traces, 0.2 mV for the zoomed insets. **c** EcoG signatures during behavioral seizures are synchronous between the ipsilateral motor cortex (blue) and central cerebellum (red). This pattern is also consistent between optogenetic (above) and KA (below)-evoked seizures.

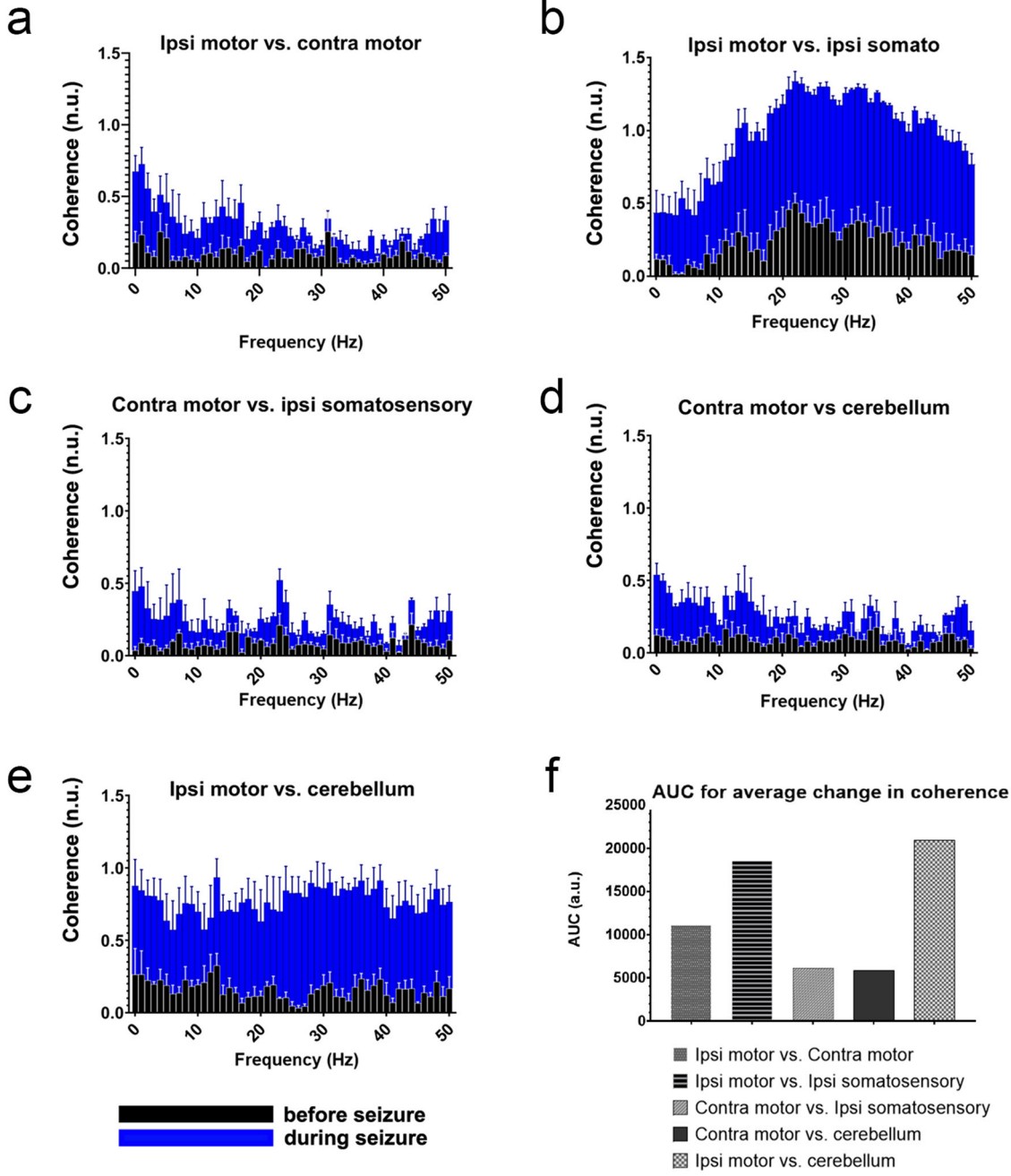

**Fig. 4 Coherence between multiple brain regions demonstrates the greatest changes between ipsilateral motor cortex and cerebellum during seizure activity. a–e** Averaged coherence ($n = 3$ biologically independent mice) from 0–50 Hz show the changes in regional coherence at baseline before seizure activity (black bars) versus during seizure activity (blue bars). The data incorporates 15 s periods of coherence immediately before light delivery (pre-seizure, black bars) and compared them to the last 15 s of stimulation to ensure the seizure was fully developed (during seizure, blue bars). Error bars represent SEM. **f** Area under the curve for average change in coherence (averaged among 3 mice) demonstrates the greatest increase in coherence during seizures between the ipsilateral somatosensory cortex/ipsilateral motor cortex and the ipsilateral motor cortex/cerebellum. *Note: statistics were not run on graph (**f**), as these reflect the average (single value) percent differences in coherence. For this reason, individual datapoints are not shown.

into the pathophysiology that eventually drive the abnormal motor behavior that is indicative of seizures.

**The cerebellum and VPM express highly coherent EcoG activity during seizures**. The cerebellum is typically used as a location of reference for EEG signal collection, an idea based on data showing that the cerebellum exhibits minimal EEG changes during seizure episodes[34]. However, many studies also observe cerebellar changes in EEG[35,36] and single-units during seizure

episodes[37–39]. In support of those studies, we found elevated cerebellar EcoG activity that was highly coherent with the ipsilateral motor cortex, specifically during seizures occurring between 13 Hz and 50 Hz (Fig. 4, $n = 3$ biologically independent mice). Of the regions studied here, the cerebellum exhibited the greatest increase in coherence with the ipsilateral cerebral hemispheres when comparing before-seizure to during-seizure periods (Fig. 4f). The increase in interregional coherence involving the cerebellum was in accordance with our finding that the cerebellum expresses both high amplitude and hypersynchronous

**Table 2 Analysis of regional brain coherence from EcoG demonstrates significant differences in the ipsi motor versus.**

**Significance of EcoG-based coherence**

| Frequency | ipsi motor vs. contra motor | ipsi motor vs. ipsi somato | Contra motor vs ipsi somato | Contra motor vs cerebellum | ipsi motor vs cerebellum |
|---|---|---|---|---|---|
| 0 | Significant | Not significant | Not significant | Not significant | Not significant |
| 1 | Significant | Not significant | Not significant | Not significant | Not significant |
| 2 | Not significant | Significant | Not significant | Not significant | Not significant |
| 3 | Not significant | Not significant | Not significant | Not significant | Not significant |
| 4 | Not significant | Not significant | Not significant | Not significant | Not significant |
| 5 | Not significant | Not significant | Not significant | Not significant | Not significant |
| 6 | Not significant | Not significant | Not significant | Not significant | Not significant |
| 7 | Not significant | Not significant | Not significant | Not significant | Not significant |
| 8 | Not significant | Not significant | Not significant | Not significant | Not significant |
| 9 | Not significant | Not significant | Not significant | Not significant | Not significant |
| 10 | Not significant | Not significant | Not significant | Not significant | Not significant |
| 11 | Not significant | Not significant | Not significant | Not significant | Not significant |
| 12 | Not significant | Significant | Not significant | Not significant | Not significant |
| 13 | Not significant | Not significant | Not significant | Not significant | Significant |
| 14 | Not significant | Not significant | Not significant | Not significant | Significant |
| 15 | Not significant | Significant | Not significant | Not significant | Not significant |
| 16 | Not significant | Significant | Not significant | Not significant | Not significant |
| 17 | Not significant | Significant | Not significant | Not significant | Significant |
| 18 | Not significant | Significant | Not significant | Not significant | Significant |
| 19 | Not significant | Significant | Not significant | Not significant | Not significant |
| 20 | Not significant | Significant | Not significant | Not significant | Not significant |
| 21 | Not significant | Significant | Not significant | Not significant | Not significant |
| 22 | Not significant | Significant | Not significant | Not significant | Not significant |
| 23 | Not significant | Significant | Not significant | Not significant | Not significant |
| 24 | Not significant | Significant | Not significant | Not significant | Significant |
| 25 | Not significant | Significant | Not significant | Not significant | Significant |
| 26 | Not significant | Not significant | Not significant | Not significant | Significant |
| 27 | Not significant | Significant | Not significant | Not significant | Significant |
| 28 | Not significant | Significant | Not significant | Not significant | Significant |
| 29 | Not significant | Significant | Not significant | Not significant | Not significant |
| 30 | Not significant | Significant | Not significant | Not significant | Not significant |
| 31 | Not significant | Significant | Not significant | Not significant | Not significant |
| 32 | Not significant | Significant | Not significant | Not significant | Significant |
| 33 | Not significant | Significant | Not significant | Not significant | Significant |
| 34 | Not significant | Significant | Not significant | Not significant | Significant |
| 35 | Not significant | Not significant | Not significant | Not significant | Not significant |
| 36 | Not significant | Not significant | Not significant | Not significant | Not significant |
| 37 | Not significant | Not significant | Not significant | Not significant | Not significant |
| 38 | Not significant | Significant | Not significant | Not significant | Not significant |
| 39 | Not significant | Significant | Not significant | Not significant | Significant |
| 40 | Not significant | Significant | Not significant | Not significant | Not significant |
| 41 | Not significant | Significant | Not significant | Not significant | Not significant |
| 42 | Not significant | Significant | Not significant | Not significant | Not significant |
| 43 | Significant | Significant | Not significant | Not significant | Significant |
| 44 | Not significant | Significant | Not significant | Not significant | Significant |
| 45 | Not significant | Significant | Not significant | Not significant | Significant |
| 46 | Not significant | Significant | Not significant | Not significant | Not significant |
| 47 | Not significant | Significant | Not significant | Not significant | Significant |
| 48 | Not significant | Significant | Not significant | Not significant | Not significant |
| 49 | Not significant | Significant | Not significant | Not significant | Significant |
| 50 | Not significant | Not significant | Not significant | Not significant | Significant |

ipsi somatosensory cortices and in the ipsi motor cortex versus the cerebellum. Paired t-tests were used to compare each frequency before versus during stimulation. Significance was set at $p$ values < 0.05 ($n = 3$ biologically independent mice).

activity during optogenetic-induced seizures that were triggered by VPM activation. These data suggest that the cerebellum is highly active during generalized motor seizures.

**Optogenetic stimulation of input to the VPM generates a dynamic, heterogeneous firing pattern in the connected thalamic neurons.** We next sought to determine the electrophysiological

properties that drive optogenetic-induced seizures in the *Ntsr1^Cre^;ChR2-EYFP* mice. Specifically, we asked how light stimulation of axons projecting to the VPM affect resident thalamic neuronal activity to initiate seizure behavior. A caveat of optogenetics, however, is that the initial neural activity measured is a direct consequence of our light paradigm: therefore, the first few seconds of activity may reflect an artificial, exogenously-induced pattern in response to light stimulation rather than a

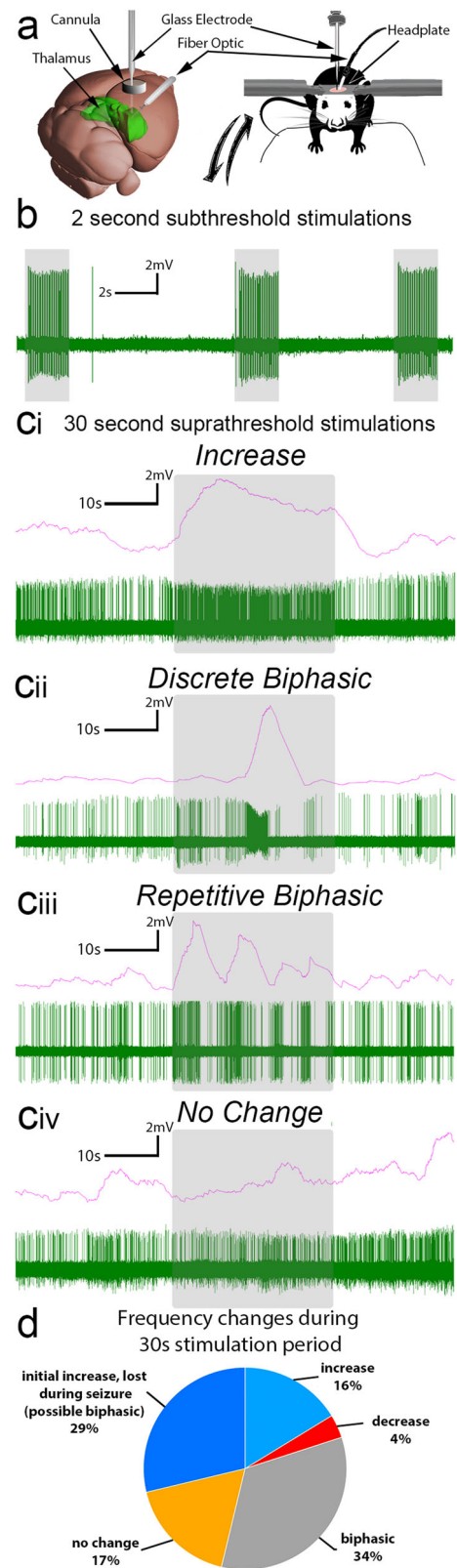

**Fig. 5 Single cell activity is altered in the VPM during seizure activity.**
**a** Schematic of the optogenetic setup. We targeted the VPM (thalamus, green) with a fiber implanted at an angle and create a craniotomy above the cortex to access the VPM with a pulled glass electrode for recordings. The mouse itself remains awake and is head-fixed on a rotating wheel, which allows for movement during recordings. Anatomical schematic of the mouse brain was adapted from Allen Brain Atlas's Brain Explorer 2. **b** Short, subthreshold-seizure stimulations (2 s light pulses at 30 Hz, gray overlays) elicit reliable entraining of VPM neurons characterized by strict excitation. **c** Delivery of seizure-inducing trains of 30 Hz light (30 s light pulses, gray overlays) elicits heterogeneous responses in thalamic neurons including (**ci**) excitation, (**cii**) discreet biphasic, (**ciii**) repetitive biphasic, and (**civ**) no change in activity during seizures. Red traces demonstrate mean frequency over 2 s bins. **d** Quantification of neural responses to seizure behavior. Only suprathreshold (seizure-inducing) stimulations are included, $n = 5$ biologically independent mice, 71 cells.

signals, the subsequent activity may indeed reflect neuronal activity that participates in the resulting seizures.

Here, we used glass electrodes to record single-unit activity in the thalamus of awake, behaving $Ntsr1^{Cre};ChR2\text{-}EYFP$ mice implanted with fibers targeted to the ipsilateral VPM. To achieve the necessary stability for recordings, mice were secured over a freely rotating wheel with a headplate and were habituated for 30 min prior to experimentation (Fig. 5a). While we estimate the regional identity of neurons based upon stereotaxic coordinates as well as post hoc examination of the electrode track through the tissue, the exact anatomical location is difficult to concretely pinpoint, as the tissue does not clearly demarcate our sharp electrode tips. We therefore anticipate that some of the recorded cells could have been sampled from beyond the limits of the VPM. However, due to the technically challenging nature of this technique, we included all the cells that we recorded from the thalamus. In addition, we reasoned that having as much single-unit information as possible would be informative. Using this approach, we successfully recorded the in vivo activity of more than 70 thalamic neurons ($n = 5$ biologically independent mice).

The thalamus contains a heterogeneous mix of neurons with different intrinsic properties and responses to neuromodulators[40]. Here, we found that baseline thalamic firing (prior to optogenetic manipulation) was primarily bursty with frequencies ranging from <1 Hz to 25 Hz. We tested how these neurons responded to two different light stimulation paradigms: 2 s of 30 Hz light, in which mice did not exhibit behavioral seizures, and 30 s of 30 Hz light delivery, which induced generalized convulsive seizures during recordings. With the delivery of subthreshold, 2 s light pulses to the ipsilateral VPM (no behavioral seizures induced), thalamic neurons expectedly demonstrated an excitatory response, indicating that blue light stimulation of the incoming ChR2-expressing axons resulted primarily in excitatory synapses onto thalamic cells (Fig. 5b). Of the 16 neurons recorded in the *subthreshold* group, 14 exhibited a significant increase in frequency when stimulated. The remaining 2 neurons did not show a significant change in frequency: it is likely that these neurons represent cells that are not innervated by ChR2-expressing fibers and are therefore largely independent of the light stimulus. With these short 2 s light stimulations, changes in firing frequencies were not sustained, as no significant difference was found between the frequency of thalamic activity 20 s before stimulation and 5 s after the end of our 2 s stimulation ($n = 15$ cells, paired t-test, $p = 0.1479$, Supplementary Fig. 4a).

In contrast to subthreshold stimulations, we found that neuronal activity underlying a 30 s seizure-inducing light stimulus was less predictable: in addition to excitation

spontaneous, endogenous pattern of activity. Nonetheless, it is important to consider that these artificial patterns of activity are sufficient to induce behavioral seizures; therefore, with the understanding that the optogenetic approach is purely a circuit model, we unveil the types of activity patterns that potentially drive seizure-related activity. In other words, while the first few seconds of activity may not be representative of natural biological

(Fig. 5ci), which may be a more direct readout of light activation than of seizure activity, they also exhibited various biphasic response profiles (Fig. 5cii–ciii) or were unaffected by the light stimulation (Fig. 5civ). When selecting only those cells in which we maintained recordings of >20 s of post-stimulation activity, we noted that most cells returned to their baseline activity within 20 s ($n = 39$ cells, paired t-test, $p = 0.110$, Supplementary Fig. 4b). To determine whether response type could be predicted by the original firing frequency, we then plotted the firing frequencies of neurons before stimulation according to their activity changes (we classified the cells as no change $n = 7$, increasers $n = 9$, biphasics $n = 20$, decreasers $n = 3$, total $n = 39$ cells). On average, there was no statistical difference between baseline frequency and how the neurons responded to the stimulation paradigm (one-way ANOVA with Tukey's multiple comparisons test, Supplementary Fig. 4c).

The number of neurons in each response class during suprathreshold stimulation—no change, increase, biphasic, and decrease—was not uniform. Most resident neurons (83%) exhibited some degree of firing response: only 17% were unaffected by light stimulation (Fig. 5d). We postulate that these non-responsive neurons represent a class of interneurons that do not engage in hypersynchrony or are not innervated by *Cre*-expressing axons. Interestingly, only 16% of the recorded neurons exhibited a strict increase in frequency. Most cells engaged in a biphasic response, alternating between periods of rapid firing and silence. Due to a high concentration of T-type calcium currents, neurons in the VPM are known to rebound with bursts of excitation following release from hyperpolarization[41]. Therefore, during a seizure, thalamic neurons may utilize their discrete properties to transition between states of inhibition and excitation.

**Injection of a Cre-dependent virus into the VPM reveals the sources of stimulated axons.** Based on the hypothesis that the VPM is a strong seizure locus, we next asked which of the *Cre*-expressing brain regions that project to the VPM is capable of driving the changes in thalamic activity and eliciting this robust seizure phenotype. We therefore injected rAAV2-retro-Ef1a-DIO-mCherry-WPRE into the VPM of *Ntsr1*[Cre] mice (Fig. 6a, $n = 5$ biologically independent mice). In this experiment, only *Cre*-positive neurons projecting directly to VPM recombine to express mCherry: thus, the use of this virus revealed the specific inputs to the VPM that we modulate in our experiments. After allowing 5 weeks for viral tracing and recombination, mice were perfused and prepared for immunohistochemistry. Despite the presence of multiple *Cre*-expressing regions in *Ntsr1*[Cre] animals, only layers 5/6 of the cerebral cortex and cerebellar nuclei demonstrated direct projections to the VPM after virus labeling. The most robust axonal bundles filled by virus injection originated from the somatosensory areas and cerebellar nuclei (Fig. 6b–g). We did not find cell body labeling from other areas that are known to project to the VPM, including the insular cortex, reticular thalamic nucleus, internal segment of the globus pallidus, and amygdala, although these areas did exhibit some mCherry fluorescence as puncta or fibers stemming from other regions that were marked during the electrode penetration (Fig. 6d). We therefore next explored the contribution of the cerebral cortex and cerebellum to seizure generation.

**Drug-induced inhibition of the cerebellum prevents the induction of seizures from the VPM.** Based on virus labeling of the cerebral cortex and cerebellum and that these regions mediate a strong influence on VPM-induced seizure activity, we asked whether the cerebral cortex and/or cerebellar nuclei are necessary

for initiating the phenotype. To parse apart the relative contribution of each region, we used lidocaine to reversibly block activity in each. While it would be interesting to specifically block the VPM itself, implanting both optic fibers and relatively large drug delivery cannulas to the same region is spatially challenging. Therefore, we focused on blocking the sources of input to the thalamic regions. Due to the rapid kinetics of lidocaine and the reversibility of its effects, we performed within-trial comparisons of seizure induction before administration, immediately after lidocaine delivery, and after lidocaine washout. We utilized our original stimulation paradigm—1 h daily stimulations of 30 s light-on followed by 60 s light-off—to ensure elicitation of a robust seizure prior to lidocaine administration. Only animals that demonstrated at least a stage 3 seizure (facial and forelimb clonus with lordic posture, with the rating experimenter blinded to the condition) with stimulation were used for these experiments. Lidocaine was administered 30 min after the first evoked seizure was resolved in order to ensure full recovery and reset of brain circuits. 5 min thereafter, animals were tested for seizure activity by applying a 30 s train of 30 Hz light to the VPM ("during" lidocaine). After 1 h from initial lidocaine delivery, mice were stimulated once again with a 30 s, 30 Hz light pulse.

Lidocaine (4%) was delivered through implanted cannulas that bilaterally targeted the cerebellar nuclei (Fig. 7a, c), somatosensory cortex, and motor cortices (Fig. 7b). Based on previous studies, we calculated that the neural inactivation window would extend ~5–30 min after initial delivery[42,43]. Furthermore, previous work from our lab used methylene blue and an anti-lidocaine antibody to detect the spread of lidocaine and determined that the drug can travel approximately 1 mm from the injection site[44]. In these studies, White et al. determined that this degree of drug spread within the cerebellar output nuclei is sufficient to result in behavioral rescue of dystonia[44]; however, the anatomical layout and size of both somatosensory and motor cortices present a challenge for a comparable and complete drug-induced inhibition of these regions. Moreover, targeting either cortical area in its entirety would be impractical: we therefore delivered lidocaine specifically to facial and forelimb regions. We then evaluated whether silencing functional connectivity there interrupted the ability to elicit subsequent seizures. Because we found that lidocaine administration yielded somewhat of an all or nothing response rather than graded changes in seizures, we rated improvement on a binary scale: that is, seizure or no seizure. Each mouse served as its own control: "before" drug delivery and "after washout" stimulations were directly compared to stimulations applied shortly after lidocaine administration—in this regard, we were confident that any changes observed were due to the lidocaine administration and not from an artifact such as tissue heating.

We found that pharmacological inhibition of the cerebellar nuclei (Dunnett's multiple comparisons test, $f = 73.53$, $p < 0.0001$), but not somatosensory ($f = 0.2170$, $p = 0.9562$) or motor cortical regions ($f = 0.1291$, $p = 0.8835$), eliminated seizure induction (Fig. 7d, Supplementary Video 4, $n = 5$ biologically independent samples for cerebellar silencing, $n = 3$ biologically independent samples per cortex silencing). Following lidocaine washout from the cerebellum, the ability to induce a seizure was rescued. Although microinfusions to the cortices did not appear to change seizure severity, unlike the dramatic improvement with cerebellar lidocaine, it is possible that the cortex contains internal mechanisms in place to compensate for isolated regions of disruption. Alternatively, the volume of lidocaine sufficient for cerebellar block may be insufficient for inhibiting the cerebral cortices. Curiously, in one mouse where excess lidocaine was delivered to the somatosensory cortex, the animal developed spontaneous seizures every 1–2 min without optogenetic

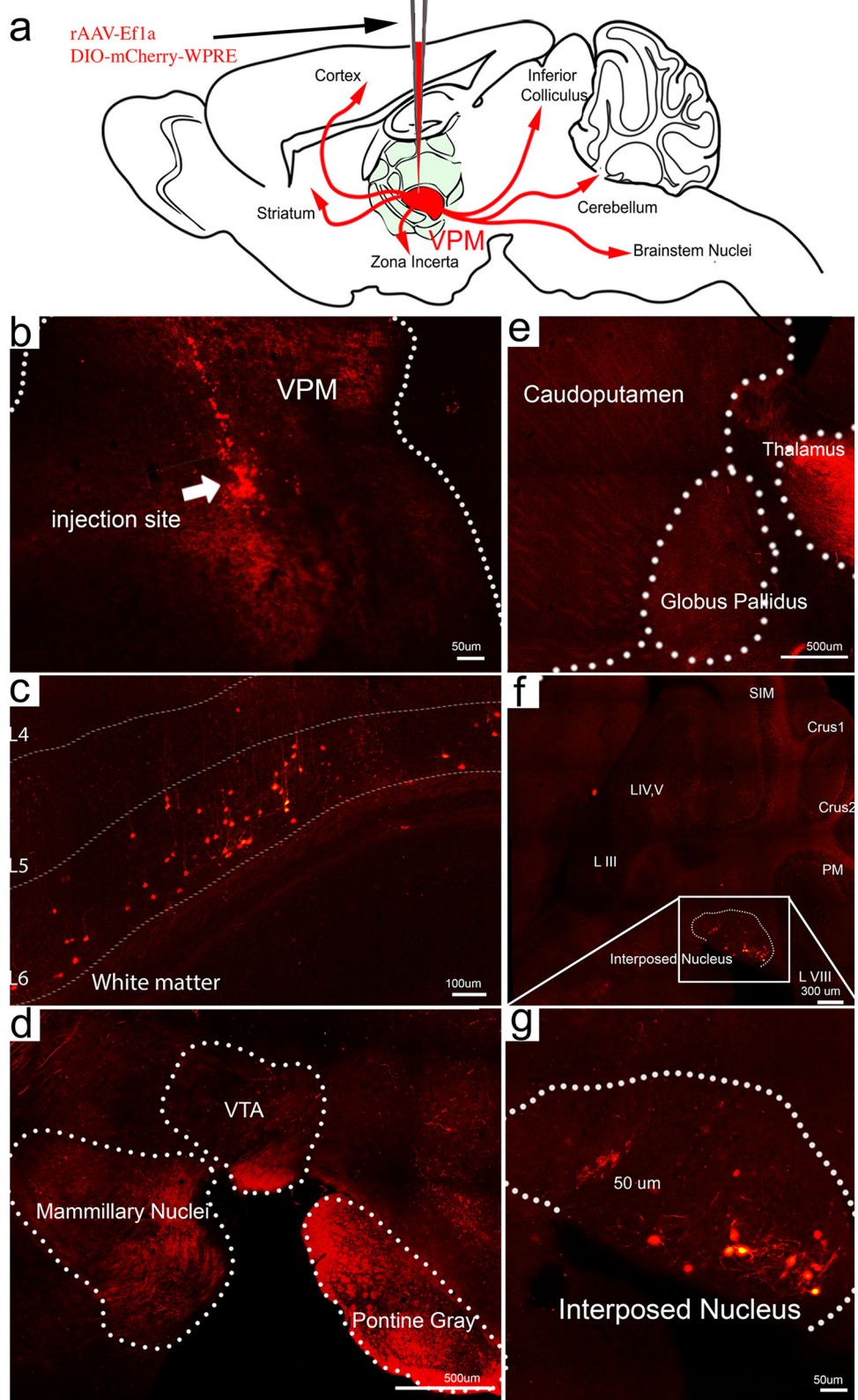

**Fig. 6 Cre-dependent viral retrograde tracing from the VPM reveals input primarily from the cerebral cortex and cerebellar nuclei. a** Schematic of the experimental setup. *Ntsr1^{Cre}* mice received unilateral injections with 680 nL of rAAV-Ef1a-DIO-mCherry-WPRE into the VPM of the thalamus. Virus taken up by local VPM neurons (virus injection site, **b**) was monosynaptically traced in the retrograde direction to the source neurons that send the projections. In these mice, only the cerebral cortex (**c**, scale bar = 100 μm) and cerebellar nuclei (**f**, **g**, scale bar = 50 μm) demonstrated a strong signal after the injections. Examples of areas that had Cre-induced reporter expression in the TdTomato mice but not in our virus injections include the VTA (**d**, scale bar = 500 μm) and striatum (**e**, scale bar = 500 μm). In areas like the pontine gray and mamillary nuclei, virus is visible as fiber tracts (or speckled background) but not in the resident nuclei (**d**).

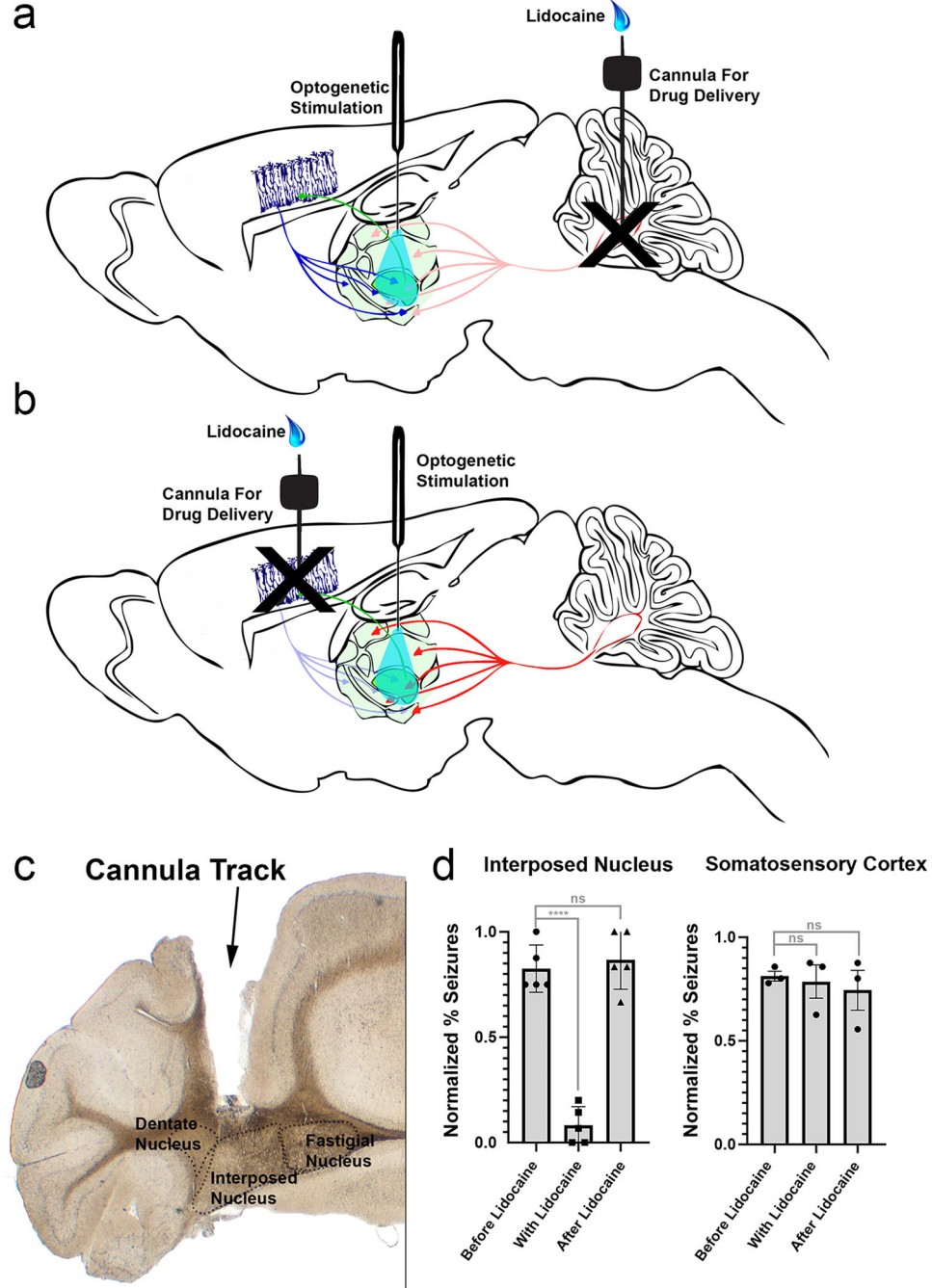

**Fig. 7 Pharmacological inactivation of individual Cre-expression inputs to the thalamus reveals the requirement of the cerebellum for seizure initiation. a** Schematic of the experimental setup for cerebellar silencing includes a surgically implanted fiber optic targeted to the VPM and bilateral cannulas targeted to the interposed cerebellar nuclei for the purpose of lidocaine delivery. **b** Schematic of the experimental setup for cerebral cortical pharmacological silencing includes a surgically implanted fiber optic targeted to the VPM and bilateral cannulas targeted to the somatosensory or motor cortices for lidocaine delivery. **c** Example of a cannula targeting to the interposed cerebellar nucleus in a 40 μm coronal slice. **d** Quantifications of scored seizures after providing trains of 30 Hz blue light pulses to the VPM before lidocaine delivery, immediately following delivery, and after lidocaine washout. A significant decrease in ability to elicit seizures occurred following cerebellar interposed nucleus silencing (Dunnett's multiple comparisons test, $p < 0.0001$, $n = 5$ biologically independent mice), but not somatosensory cortex ($p = 0.9562$) or motor cortical ($p = 0.8835$) silencing ($n = 3$ biologically independent mice per cortical condition). Error bars represent SEM.

interference until it was euthanized. Together, these data support the hypothesis that cerebellar connectivity to the VPM is essential for seizure initiation in this optogenetic model, whereas specific regions of the cerebral cortex may be more heavily involved in generalizing seizure activity after it has been initiated and processed elsewhere in the brain (the thalamus).

## Discussion

In this study, we devised an optogenetic mouse model to determine the role of the VPM in the generation of seizures. Recent use of optogenetics to induce seizures shows that these models provide numerous advantages over conventional models: primarily, they provide genetic control over the identified neurons,

an ideal feature for disentangling seizure dynamics, and can reliably initiate seizures within a predictable time window[45–47]. While optogenetic induction may not reflect all cases of seizures, which can arise from multiple brain regions or pathologies, it does minimize the confounding variables present in chemo- and electro-convulsant models, as light neither acts upon unknown ligands nor is sustained following termination. For our model, we used cell-type specificity to modulate a genetically defined pool of neurons that evokes seizures in otherwise normal mice and then we tracked the spread of abnormal brain activity. Using an *Ntsr1*[Cre] transgene to drive the expression of a light-sensitive channel in major seizure-engaged regions, we tested the anatomical, neuronal, and circuit properties of seizure induction. We anticipate the value of this model in additional studies, including screening of anti-epileptic drugs, due to its simplicity and normal development.

A reasonable assumption in seizure models is that co-activation of multiple circuits, regardless of their specific identities, could elicit a seizure if such activation surpassed a particular threshold. This would be especially relevant to the thalamus, as its interconnectivity with the rest of the brain is unparalleled. Surprisingly, stimulation of functionally similar, anatomically adjacent, and equally innervated regions as the VPM in our *Ntsr1*[Cre] animals did not elicit behavioral changes, suggesting that the identity of circuits involved is crucial. It is especially pertinent, therefore, to utilize such optogenetic models in the future to parse apart ictogenic circuits from those recruited only after initiation.

Neurons within the thalamic nuclei have distinct genetic profiles, morphologies, firing patterns, and synaptic properties[48]. Despite the canonical view of homogenous, synchronous activity during seizures, calcium imaging and multiarray studies indicate the presence of heterogenous activity among neuronal populations both at seizure onset as well as throughout the seizure period[49–55]. Furthermore, numerous studies in animal models have established that a proportion of cells within the epileptic brain remain largely normal during seizure episodes. For example, in a cat model of focal epilepsy, only 30% of recorded neurons demonstrated an epileptic response[50]. In humans and monkeys, as many as 51% of neurons recorded by multielectrode arrays were non-responsive during seizures[55]. Here, we demonstrate that perhaps a higher proportion—as many as 80%—of VPM neurons may change their activity and actively contribute to initiating or sustaining seizures. Within this population, the activity of the affected neurons can take on different forms, altogether suggesting a high level of heterogeneity in neural responses as well as the identity of neurons that respond versus those that remain unchanged. Whether this is true for other regions of the brain has yet to be resolved.

Consistent with the presence of heterogeneous activity, there is now evidence that instead of excitatory neurons being exclusively responsible for initiating seizures, an increase in inhibitory interneuron activity may first cause local populations of neurons to become quiescent, only to subsequently rebound with sudden bursts of sustained activity. It has been suggested that parvbumin-expressing interneurons may be particularly important for this form of ictogenesis[56]. The VPM is known to target both excitatory cortical pyramidal cells as well as GABAergic parvalbumin interneurons[57–59]. In fact, in an investigative long-range tracing study, the authors determined that the VPM accounts for 72% of subcortical input to the PV neurons[60]. In addition, PV neurons within the reticular thalamic nucleus were reciprocally connected to the VPM[59]. It is possible, therefore, that excitatory-inhibitory neuron interconnectivity between the VPM and surrounding regions partially account for the specific seizure activity observed in our optogenetic induction model.

Our data demonstrate that the cerebellum may engage in chronic network hyperactivity during seizures, which supports a growing literature suggesting a dynamic role of the cerebellum in epilepsy. The sharp increase in coherence between the cerebellum and the ipsilateral motor cortex, the EcoG location closest to VPM seizure onset, underscores the influence of the cerebellum in seizure activity, especially given the lack of coherence between these regions prior to stimulation (Fig. 4e). This analysis does not, however, exclude the possibility of antidromic activation of the cerebellum, which may partially account for its engagement. Although a different paradigm than optogenetics, we previously established that low frequency deep brain stimulation (0–13 Hz) to the interposed nucleus alleviates ataxia, whereas the same stimulation protocol provided in the absence of input from upstream Purkinje cells does not result in recovery[61]; thus, retrograde cerebellar effects can indeed have implications on downstream circuits such as those integrated with the VPM. It is possible that in addition to the optogenetic activation of anterograde signals in the stimulated axons and terminals within the VPM, the same stimulated axons could send induced retrograde signals that travel back into the cerebellar nuclei. From there, they could loop through the cerebellar cortex and exit via the same tract, or because at least some cerebellar nuclei projections bifurcate to innervate multiple regions[62], the induced signal could ultimately impact more than just the VPM. Moreover, en passant axons could also be activated in our stimulation paradigm. However, it is not known whether the strength of these alternate pathways would provide the same influence on the circuit and behavior as the anterograde activity that we examined by electrophysiology near the source of light in the VPM (Fig. 5). In either scenario, high coherence between the cerebellum and cerebral cortex may not be completely unexpected: nonetheless, when considered in conjunction with the lidocaine blocking experiments, it is likely that cerebellar activity plays an active role with the VPM in inducing seizures (regardless of the route that takes to get there).

We manipulated the optogenetic response by delivering lidocaine to the cerebellar nuclei to eliminate its functional output. Without this connectivity, seizure induction was blocked. Based on evidence of minimal lidocaine spread[44], we predicted its effect to remain local to within the cerebellar nuclei—however, there is a body of literature suggesting that the binding of lidocaine to sodium channels can occur from within those neurons, which allows for its silencing action to take effect throughout the entire affected neuron, even at downstream regions[63,64]. Therefore, it can be expected that even with stimulation of axons within the VPM, those that stem from cerebellum have sodium channel blockage even at their distal sites within the thalamus. We interpret the results with the possibility that a portion of cerebellar axons are silenced, although some cerebellar axons could remain active, albeit with their cell bodies blocked.

Contrarily, if assuming that cerebellar cell bodies were sufficiently blocked by lidocaine but their distal axons remained active when stimulated in the thalamus, the observation of seizure eradication would indicate that back propagation of the optogenetic signal to cerebellar nuclei is necessary for induction. In other words, it is possible that cerebellar projections to other regions outside of the thalamus are also required for the seizure phenotype. Alternatively, assuming that the cerebellar nuclei cell bodies and their distal axons were blocked by lidocaine administration, the results would indicate that the specific cerebellar pathway involved in seizure initiation is the cerebellum to VPM projection. Regardless, this experiment suggests that the cerebellum is one component of a larger seizure circuit.

Despite literature and our own work suggesting that abnormal cerebellar activity drives seizures, there is also evidence proposing

the opposite: that decreasing cerebellar activity worsens seizures. For example, clinical observations since the early 1800's have noted cerebellar atrophy in patients with prolonged seizures; immunohistochemistry experiments report lower dFosB expression in cerebellar nuclei following seizures, indicating a decrease in neuronal activity[65]; pharmacological inhibition of the cerebellar nuclei increased abnormal EcoG activity in absence mouse models, whereas excitation blocked their occurance[66]. Despite these seemingly contradictory results, both scenarios could co-exist. Depending on the pattern of cerebellar activity, it could either synchronize or disrupt downstream networks to modulate seizures. In genetic models of epilepsy, cerebellar firing is abnormal[39,65,67,68]. During periods of absence seizures, the cerebellum switches from rhythmic to asynchronous firing[66,68]. This rhythm of cerebellar output may force synchrony between regions like the thalamus and cerebral cortex. In these disease models, where baseline cerebellar activity is already disrupted, delivery of stimulation through closed-loop DBS or optogenetic pulses of light may disrupt synchronized networks and terminate seizures. In fact, a recent study blocked seizures with specific DBS parameters targeted to the cerebellar nuclei[69]. However, in models with normal baseline cerebellar activity, delivery of a patterned train of light to cerebellar nuclei instead forces the same bursty pattern that is inherent in epilepsy models, thereby initiating synchrony in upstream regions to cause a seizure. In patients with genetic forms of epilepsy, the cerebellum may therefore be an important target for treatment to correct abnormal output. In idiopathic or cryptogenic epilepsies, which make up an overwhelming 65% of cases, abnormalities in the cerebellum may be at fault for the spontaneous emergence of seizures. In both cases, the cerebellum should be considered as a key player in epilepsy pathophysiology.

In these studies, we demonstrate that convergent input to a facial region of the thalamus, the VPM, is a source of generalized motor seizures. Specifically, 30 Hz co-activation of cerebral cortical and cerebellar inputs to the VPM induce these severe seizures. With virus tracing and reversible inactivation, we show that of these circuit inputs, the cerebellum has a predominant role in seizure initiation, supporting recent work that links cerebellar connectivity to seizures. Furthermore, using single-unit in vivo electrophysiology recordings performed in seizing mice, we uncover heterogeneous, biphasic activity of thalamic neurons during seizures. Together, these data suggest that the cerebellum likely plays a critical role in abnormal signaling at seizure onset. We therefore propose the cerebello-thalamic circuit as a putative target for seizure treatment.

## Methods

**Animals**. All relevant ethical regulations for animal testing were complied with for subsequent studies. All mice were housed in an AALAS-accredited facility on a 14 h light cycle. Husbandry, housing, euthanasia, and experimental guidelines were reviewed and approved by the Institutional Animal Care and Use Committee (IACUC) of Baylor College of Medicine (protocol number: AN-5996). To develop transgenic mice with restricted TdTomato or Sun1 expression, we crossed heterozygous Ntsr1[Cre] males (B6.FVB(Cg)-Tg(Ntsr1-cre)GN220Gsat/Mmucd, University of California, Davis MMRRC, Mouse Biology Program) to het- or homozygous ROSA[lox-stop-lox-TdTomato] or ROSA[lox-stop-lox-Sun1]λlines (B6;129-Gt(ROSA)26Sor[tm5(CAG-Sun1/sfGFP)Nat]/J, Jackson Laboratory, Bar Harbor, Maine). These offspring, whose genotype was confirmed by PCR of digested tissue (tail snips or ear punches), were used for immunohistochemistry. To gain light-sensitive control over these neurons, we crossed Ntsr1[Cre] males to ROSA[lox-stop-lox-ChR2-EYFP] (B6;129-Gt(ROSA)26Sor[tm32(CAG-COP4*H134R/EYFP)Hze]/J, Jackson Laboratory, Bar Harbor, Maine) females. Again, their genotypes were confirmed through PCR of tail or ear tissue, but also confirmed by presence of ChR2-EYFP signal in select neurons (it is important to ensure that ChR2 is not present in all neurons, which would indicate germline transmission). For experimental studies, the day of birth was considered as postnatal day 0 (P0), which was a useful time marker for accurately and consistently aging the mice to between P60 to P180, which was the age range used for the experiments. Mice of both sexes were used equally, as no sex differences were observed throughout experiments. For instance, we compared the

seizure ratings from 4 male and 4 female mice: using a t-test to examine the male/female data, we did not find a significant difference (at $p = 0.1955011095$). Opsin or Cre-negative littermates were used as non-expressing seizure controls. The mice were housed on a 14 h/10 h light/dark cycle. All animal studies were carried out under an approved institutional animal care and use committee animal protocol according to the institutional guidelines at the Baylor College of Medicine.

**Surgery**. Thirty minutes prior to any surgical procedure, mice were administered pre-operative buprenorphine (0.6 mg/kg subcutaneous) and meloxicam (4 mg/kg subcutaneous) as analgesics. Surgery for awake optogenetics was performed as detailed in Ung and Arenkiel, 2012[7]. Briefly, mice were anesthetized with 4% isoflurane in an induction chamber until unresponsive to the toe-pinch reflex. After transferring to a stereotaxic apparatus, isoflurane was lowered and maintained at 2% for the duration of surgery. Using aseptic technique and a sterile work field, the mouse was cleaned and prepped, the skin above the skull opened, and a small craniotomy of approximately 1 mm in diameter made above the VPM (AP −1.6 mm, ML +/−1.55 mm). A polished optic fiber previously glued into a ceramic ferrule (Thorlabs, Newton, NJ, USA; #CFLC230-10) was then gently lowered to the approximate depth (2.8 mm) and secured with Metabond Adhesive Luting Cement. For most surgeries in which EcoG was also incorporated, 6 additional craniotomies were made above the superior colliculi (reference and ground electrodes, AP −5.21 mm, ML +/−1 mm), primary sensory cortex (AP + 1 mm, ML + 3.5 mm), primary motor cortices (AP −1 mm, ML +/−1 mm), and cerebellum (AP −6 mm, midline). An EcoG head mount compatible with a detachable preamplifier (Pinnacle Technology, Inc, Lawrence, KS, USA; #8406) was equipped with six silver ball-tipped wires (A-M Systems, Sequim, WA, USA; #785500) soldered to individual pins. This head mount was glued to the most rostral portion of the skull. The wire free ends were inserted into the craniotomies between the skull and dura. All wires were secured with UV epoxy followed by Metabond Adhesive Luting Cement. Next, the fiber optic implant as well as an EcoG head mount were wrapped in dental cement and cement borders secured with Elmer's Instant Krazy Glue.

**Post-operative care**. After all surgeries, mice were placed in a warming box (V500, Peco Services Ltd., Cumbria, UK) to prevent hypothermia while the anesthesia wore off. Once they were awake and mobile, they were returned to the home cage. As part of the post-operative care regime, the mice were provided with buprenorphine (0.6 mg/kg subcutaneous) every 12 h for at least 72 h. In addition, the mice are also administered Meloxicam at 4 mg/kg (subcutaneous) every 24 h for the 72-h duration. Mice were carefully monitored and allowed to recover for 2–3 days before optogenetic stimulation and recordings were initiated, and 5 weeks before viral expression was analyzed.

**Optogenetic seizure induction and EcoG recordings**. To induce seizures in this mouse model post-surgery, a fiber optic cable powered by a 465 nm LED light box (ALA Scientific Instruments Inc, Farmingdale, NY, USA) was carefully inserted into a mating sleeve attached to the unilateral fiber implant. When EcoG was included, a custom-made preamplifier (Pinnacle Technology, Inc, Lawrence, KS, USA; #8406) was connected to the implanted EcoG head mount. All EcoG and optogenetic stimulation parameters with subsequent data was recorded through Spike2 software and delivered using a CED Power1401 data acquisition interface (CED, Cambridge, UK). Recordings were paired with live video acquisition and were monitored for 5 min before stimulation, the duration of stimulation, and 5 min after stimulation. To entrain the circuits to seizures, 1 h of 30 Hz trains of optogenetic stimulation, 30 s light on and 60 s lights off, was applied daily in the early afternoon for 1 week. Depending upon the precise targeting of the fiber optics, severe seizures would typically be triggered upon the first stimulation of the day beginning on day 2–3 if not on the first day. Seizures were reliably induced every day after the first successful induction. While a full 30 s period of stimulation was not necessary for inducing seizures after the first successful seizure (15 s was sufficient), 30 s trains were used for most experiments to maintain consistency. Of note, cameras with different frame rates were used, which altered the visual appearance of light patterns in some cases. However, despite the visual differences via video, all stimulations were kept consistent at 30 Hz. Maximum LED power at the end of the implanted fiber was measured to be ~3.6 mW and stimulation consisted of this maximum brightness to induce seizures.

**KA seizure induction**. Kainic acid (KA) was dissolved in 10 mM phosphate-buffered saline (PBS) and the pH was adjusted to 7.2. Mice were injected intraperitoneally with KA at 30 mg/kg and carefully monitored thereafter. If no seizure developed within 30 min, an additional 5 mg/kg dose was administered. Seizures were then rated by a blinded experimenter using a modified Racine scale[31] (Table 1) and sacrificed following the procedure.

**Tissue processing and immunohistochemistry**. For euthanasia during perfusion, mice were deeply anesthetized with fresh avertin/tribromoethanol (0.2 ml/10 g body weight, intraperitoneal of a 1.25% solution). When no toe pinch reflex was detected, the mice were perfused with 1 M phosphate-buffered saline (PBS) followed by 4% paraformaldehyde (PFA) by slowly pumping these solutions through

the left ventricle of the heart. The brain was then carefully dissected and submerged in 4% PFA for 1–2 days. Following serial sucrose protection (15–30% sucrose in PBS), samples were submerged in OCT and immediately frozen at −80 °C. For immunohistochemistry, free-floating sections cut at 40um on the cryostat were incubated in antibodies in a solution of 10% normal goat or donkey serum and 0.01% Tween-20. For DAB staining, sections were first bathed in $H_2O_2$ to quench endogenous peroxidases and then washed 3 times in PBS for 5 min each. For all stains, sections were blocked for 2 h with the blocking solution at room temperature and then incubated in primary antibodies overnight at room temperature. Sections were then washed with PBS 3 times for 5 min each and then incubated in secondary antibodies for 2 h. Sections were washed again 3 times in PBS for 5 min each and mounted onto slides. DAB slides were air-dried overnight and then dehydrated in ethanol before cover-slipping. Slides with fluorescent signal were immediately cover-slipped using FluoroGel as a medium. Antibodies used for immunohistochemistry (primary antibodies) were: calbindin (Rb and mouse (Ms), Swant, 1:10,000); NeuN (Rb, Millipore, 1:500); NFH (Ms, Covance, 1:1500); TH (Rb, Millipore, 1:500); GFP (Ch, Abcam, 1:1000); Secondary antibodies were: donkey anti-mouse, anti-rabbit or anti-guinea pig secondary antibodies conjugated to Alexa 488, 555 or 647 fluorophores (Invitrogen), all diluted to 1:1500. Photomicrographs of tissue sections were captured using Zeiss AxioCam camera mounted on a Zeiss Axio Imager.M2 microscope or on a Zeiss Axio Zoom.V16. Images of the tissue sections were acquired and analyzed using either Zeiss AxioVision software (release 4.8) or Zeiss ZEN software (2012 edition). After imaging, the raw data were imported into Adobe Photoshop CS5 and corrected for brightness and contrast levels. Schematics were drawn in Adobe Illustrator CS5 and then imported into Adobe Photoshop CS5 to contract the figure.

**In vivo electrophysiology**. For head-fixed, awake recordings, mice were implanted with custom-made headplates and a craniotomy made above the VPM. A fiber optic was also implanted at a 45-degree angle to target the VPM underneath the headplate. After 72 h of recovery, the mice were habituated for 30 min per day in a head-fixed apparatus for 3 days before recording. Neuronal spikes were recorded and categorized based on standard stereotaxic coordinates measured from bregma and on firing pattern. We examined a total of 70 units in 5 animals, and selected only those cells with consistent amplitudes to ensure any changes in activity were due to our stimulation paradigms rather than movement of the cell away from, or back to, our electrode tip. All recordings were collected using ~10–20 MΩ pulled borosilicate glass pipettes filled with 0.9% saline. The recorded signals were digitized at a sampling rate of 5000 Hz into Spike2 (CED, England) where single units were verified with principal components analysis. While neurons recorded under the 2 s light delivery condition were able to be stably recorded for over 30 s after light delivery, many neurons in the 30-s light delivery condition were lost >5 s after recording due to the induction of motor seizures. Therefore, for 30 s seizure-inducing electrophysiology analyses, we selected only those cells in which we recorded >20 s of post-stimulation activity.

**Lidocaine administration**. For lidocaine experiments, mice underwent an optogenetic-implant surgery (see above). During the surgery, bilateral (ordered as singles, implanted bilaterally) cannulas from PlasticsOne (#8IMS303T3B01, cut 3.5 mm below the pedestal) were implanted to target the interposed cerebellar nuclei (AP −6.24 mm; ML 1.5 mm; DV 2.6 mm from surface of brain), the somatosensory cortex (AP + 1.645 mm; ML 2.5 mm; DV 0.5 mm from surface of brain), or motor cortex (AP −2.12 mm; ML: 2.0 mm; DV 1.5 mm from surface of brain). Following a 3-day recovery from surgery, animals were subjected to our normal optogenetic stimulation paradigm of 1-h daily stimulations, 30 s light-on/60 s light-off, at 30 Hz. Once a full generalized seizure was induced and recorded, 4% lidocaine was delivered bilaterally through the implanted cannulas 30 min after the first seizure. Next, 5 min after lidocaine administration, animals were again stimulated with 30 s of 30 Hz 465 nm light and video recorded. After 1 h from initial lidocaine administration, animals were again delivered 30 Hz light for 30 s for the washout stimulation period and video recorded.

**Statistics and reproducibility**. Sample sizes were not determined using a priori power analysis but were based on the criteria for significance in observations or based on previous publications[44]. All data was analyzed and graphed in Prism9.1.0 and *p* values were considered significant at values less than 0.05. For differences in relative fluorescence, thalamic nuclei were segmented in Photoshop and their mean gray values, area, and integrated densities were calculated. A region of layer 6 cortex was used as reference for each slice. Data was then transferred to Prism and graphed. A Dunnett's multiple comparisons test was performed to compare the means of each group with that of the VPM region. In Fig. 2c, the dots correspond to regions targeted by the fiber optic implant as determined by the most ventral area of the electrode track in 40 µm cryostat-sectioned slices.

All EcoG data was collected in Spike2 at a sampling rate of 5000 Hz. For coherence plots constructed in MatLab (Supplementary Software 1), *before stim* was computed using the 15 s immediately prior to optogenetic stimulation; *during seizure* was computed using the last 15 s of the 30 s optogenetic stimulation train, when animals were actively showing seizure-like behavior. The

Spike2 software did not allow for export of raw numbers in coherence plots (Fig. 4a–e), so values were extrapolated using Plot Digitizer and subsequently analyzed in Prism.

Single-cell quantifications were performed in Prism using paired t-tests (Supplementary Fig. 4a, b) or a one-way ANOVA (Supplementary Fig. 4c) for statistical analyses. For lidocaine experiments, presence of seizures was visually scored as 1 (seizure) or 0 (no seizure) for a train of 30 s, 30 Hz stimulations before lidocaine administration and beginning 5 min after administration. An animal was scored as "seizure" if at least facial/forelimb clonus was reliably elicited. For washout experiments, mice were returned to their home cages and re-evaluated with optogenetic stimulation and visual scoring 1 h after initial lidocaine administration. All the seizure ratings were performed by the same individual for consistency, but they were blinded to the condition (which were evaluated post hoc via video recordings). These data were then plotted and analyzed via Dunnett's multiple comparisons test, as we compared the means of *with lidocaine* and *after lidocaine* to the *before lidocaine* condition (which served as the control), in Prism. Error bars in all graphs represent SEM.

**Reporting summary**. Further information on research design is available in the Nature Portfolio Reporting Summary linked to this article.

## Data availability
Please see "Supplementary Data 1" for data generated in the experiments presented in the current study.

## Code availability
MatLab code used for EEG coherence analyses presented in the current study are available as Supplementary Software 1.

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

## Acknowledgements

This work was supported by funds from Baylor College of Medicine (BCM) and Texas Children's Hospital. R.V.S. received support from The Hamill Foundation and the National Institutes of Neurological Disorders and Stroke (NINDS) R01NS089664, R01NS100874, and R01NS119301. R.V.S. is also supported by a grant from the Dystonia Medical Research Foundation (DMRF). R.V.S., D.H.H., and Y.L. are supported by National Institute of Mental Health (NIMH) R01MH112143. Research reported in this publication was supported by the Eunice Kennedy Shriver National Institute of Child Health & Human Development of the National Institutes of Health under Award Number P50HD103555 for use of the Cell & Tissue Pathogenesis Core and the Neuroconnectivity Core. The content is solely the responsibility of the authors and does not necessarily represent the official views of the National Institutes of Health. J.B. received support from F31NS115432.

## Author contributions

J.B., J.O.-G., and R.V.S. designed the experiments. J.B., J.O.-G., T.L., and R.V.S. performed the experiments. J.B., B.B., Y.L., D.H.H., and R.V.S. designed code for electrophysiology analysis. J.B., J.O.-G., B.B., L.E.S.L., Y.L., D.H.H., B.R.A., and R.V.S. analyzed the data. J.B. and R.V.S. wrote and edited the paper.

## Competing interests

The authors declare no competing interests.
