## [Peer Review File · Communications Biology]

Reviewers' comments:

Reviewer #1 (Remarks to the Author):

COMMSBIO-22-2042-T

This is a manuscript that utilizes meticulously executed state-of-art techniques to determine cerebellum contributes via the thalamic VPM to trigger tonic-clonic seizures. Unfortunately, the manuscript significantly suffers from lack of solid background, including previous publications on a similar topic as well as from overinterpretation of its own results. Below I will point to most pressing concerns:

- (1) Line 43-45. Here the authors clearly talk about epilepsy and patients with epilepsy, but not patients with seizures. While recurrent seizures are almost sine qua non condition of epilepsy, a seizure in a patient does not mean epilepsy with long-term treatment. About 10% of population will have a seizure during lifetime. On the other hand, probability of developing epilepsy during lifetime is slightly under 4% (1:26). The authors should look into the 2010 Academy of Medicine publication on Epilepsy.
- (2) Further, the authors need to look at the most recent ILAE classification of seizures and epilepsy to be able to provide pertinent and up to date definitions throughout the paper.
- (3) Lines 50-52. There are many published models of seizures and also epilepsy besides absence seizures.
- (4) Lines 56-57, while the authors should be praised for their effort to link thalamic nuclei to tonic-clonic seizure generation, they need to look as papers from 80s from Browning and Nelson, who were able to elicit tonic-clonic seizures (but not pure clonic with preserved righting ability) after intercollicular brain stem transection, clearly indicating that all necessary circuitry for these seizures is located in the brain-stem. Indeed, there may be plethora of additional trigger points such as cortex, thalamus, basal ganglia, etc. Some structures may serve as trigger points, other as attenuation points (principle of deep brain stimulation for epilepsy).
- (5) If the authors are focusing on tonic clonic seizures, i.e. generalized seizures with motor convulsions (please see the current definitions as per #2), they should not be introducing absence seizures (generalized non-convulsive seizure with clear participation of thalamocortical circuitry) and setting the scene for absence seizures. The authors should also look into the book "Models of Seizure and Epilepsy", 1st ed 2006, 2nd ed 2010 (Schwartzkroin, Moshe, Pitkanen etc) that will also provide very useful information on relatively recent developments while bridging the less recent information in.
- (6) VPM in humans is a major relay nucleus for somatosensory information from the face and contains third order neurons of the trigemino-thalamic tract. However, the authors mention somewhere that there is certain topography within the nucleus suggesting subcompartmentalization of the nucleus (lines 156-161). This is very worth of investigating and discussing in detail. Indeed, reciprocity of VPM in humans and mice is another question as rodents have slightly different arrangements than primates (e.g., entopeduncular nc and others).
- (7) Lines 70-71 – this is very important piece of information as in acute seizure models (PTZ, picrotoxin, bicuculline, 3-MPA etc, etc, etc)... there is, in adult subjects, a continuous progression from "absence" seizures (freezing with electrographic spindles of epileptiform activity) through clonic seizures (face and forelimb clonus with preserved righting ability, EEG Spike and wave, polyspike and wave) eventually to tonic-clonic seizures (loss of righting ability) and this should be mentioned/emphasized. Agani "Models of Seizures and Epilepsy" will provide solid background here.
- (8) Lines 79-80, the reference does not seem to match cerebellar stimulation. The authors should look up the original papers of Chkhenkeli and coll
- (9) Last paragraph of introduction contains unnecessary repetitions on cerebellar stimulation that

should be deleted.

(10) Lines 136-137 do not describe the tonic-clonic seizure phenotype

(11) Lines 168-169. Primary seizures elicited by KA are NOT tonic-clonic seizures. KA induces focal seizures first with secondary generalization to clonic seizures. Only with high doses this seizure type may progress to tonic-clonic seizures (again Models of Seizures and Epilepsy will be very helpful here).

(12) A clear question to the lines 163-178 is: If you lesion VPM, can you still induce tonic-clonic seizures using high dose KA or other model of tonic-clonic seizures? A simple experiment to be done, not even requiring animals with tagged neurons/circuits. Lines 335-350 – lidocaine in VPM would help!

(13) Lines 182-194 can be substantially shortened as EEG is a distinctive technique for positive diagnosis of epilepsy seizures.

(14) Lines 196-207, an inset in the figure with shorter time scale (to see the character of the ECoG activity) would be beneficial.

(15) Wherever the authors speak about full tonic-clonic seizures in their manuscript, they need to be sure that they have observed wild run at the onset (sometimes not present), loss of posture, tonic stretch on fore and/or hindlimbs followed by a long clonus without righting present.

(16) Lines 387 and previous, optogenetics starts also with previously healthy brain....one of the shortcomings of chemically/electrically induced seizures. Moreover, may so called "cryptogenic" cases have genetic underpinning....

(17) The discussion on cerebellum should be significantly shortened.

(18) Lines 510-512 – cerebellum can very likely bypass the connections to VPM if those are severed/disable and produce tonic-clonic seizure pattern using an alternate circuitry.

(19) Was there any sex difference in the parameters followed?

(20) Table 2 – modified Racine scale: Pinel and Rovner already in 70s modified Racine scale in a sensible way to include tonic-clonic seizures. Please find the reference (Models of Seizures and Epilepsy can help)

(21) Why was Dunnett's test used? I believe you ran Kruskal-Wallis first with post hoc Dunnett's, correct? Rationale for statistics should be present as well as detailed description how the statistics was performed should be given. Wherever ANOVA results are presented, please show actual F values, degrees of freedom, and actual p value in a common annotation.

Reviewer #2 (Remarks to the Author):

This manuscript by Beckinghausen et al. describes an elegant study in which they designed a relatively novel mouse seizure model to demonstrate the ability of the ventral posteromedial nucleus of the thalamus (VPM) to induce a tonic-clonic (TC) like seizure. The model is based on an optogenetic approach in which they drove cell-specific expression of the excitatory opsin ChR2 in VPM afferents originating from the neocortex and cerebellum, which they beautifully demonstrate. By stimulating these fibers optogenetically they induce mice to display a profound seizure that includes several, previously described behavioral phenotypes characteristic of TC. The authors also provide data describing the heterogeneity of the VPM response during seizures using in vivo electrophysiological methods. Most strikingly, the authors demonstrate that by silencing the cerebellum using lidocaine, seizure induction is blocked. Overall, these results will be of significant interest to neurologists and neuroscientists looking to understand the mechanisms of seizure generation and systems level interconnectivity between the cerebellum-thalamus-and-neocortex. The overall design of the experiments is robust. The manuscript would be greatly enhanced by addressing several concerns related to both the description of the experiments, the interpretation of results, and the accuracy of the language used throughout the text. The two most pressing concerns are the inadequate analyses of the EEG and ephys. data and the lack of clarity in the description of the design and impact of the lidocaine experiments. My major and minor concerns are included in the pdf attachment.

Reviewer #3 (Remarks to the Author):

I have reviewed the manuscript "The cerebellum contributes to tonic-clonic seizures by altering neuronal activity in the ventral posteromedial nucleus (VPM) of the thalamus", submitted to Communications Biology. This paper uses in vivo optogenetics and electrophysiology to examine whether a specific thalamic nuclei, VPM, and inputs to those neurons can cause tonic-clonic seizures. As the authors review, the role of the thalamocortical circuit is well-studied in absence epilepsy, but the role of the thalamus, and in particular its inputs from the cerebellum, are less understood in general and in other forms of epilepsy. The major finding that specifically silencing cerebellar inputs during seizure-inducing stimuli almost completely blocks the effect is intriguing.

Overall the study is well founded and the experiments are well executed. The general area in which this manuscript could be improved by revisions is in some of the interpretations of the data and in the organization of the discussion section. Below are my specific comments, but the manuscript would be improved by a careful review of the wording of the entire document to ensure that the statements made are consistent with the findings presented.

First is what seems to me the most important caveat to this study. Very briefly mentioned in the Discussion section is that "This analysis does not, however, exclude the possibility of antidromic activation of the cerebellum, which may partially account for its engagement." The discussion of this topic needs to be addressed in a much more expansive way. Activation of cerebellar nuclei axons in VPM will not only activate the cerebellar nuclei through antidromic action potentials, it will also activate all the other targets that are contacted by en passant synapses or branching axons. This is true for axons from other inputs as well. Thus, even if the light stimulation is confined to VPM it may be activating every other downstream target of the brain areas that project to VPM. This does not make the findings that blocking cerebellar inputs blocks seizures any less interesting, but the authors should be wary of describing their stimulation as specific to VPM.

Results section:

As a general comment, I think that the information found in Figure S1 is important enough to include in Figure 1. Some of the panels could be removed as the lower magnification images (Fig S1 panels D-F and J-L) do not add more information than is contained in the higher magnification images.

With the methods section pushed to the end it helps the reader to have some details included in the Results section. The statement "We found that unilateral delivery of 30Hz pulses of light to the ventral posteromedial nucleus (VPM) elicits a robust, TC seizure phenotype..." understates the nature of the stimuli by not including the context that the 30Hz light is delivered for an hour a day for multiple days.

The specificity of VPM stimulation is not very convincing (even beyond the issue of activating other targets of branching axons mentioned above). ChR2 expression is very high in neighboring nuclei (presumably from axons/terminals), and unless the authors discuss the penetration of light through this tissue it is hard to imagine that those nuclei are not also activated either by light or by antidromic activation of collaterals of these axons. Additionally, the claim that stimulating the surrounding nuclei, in particular VPL and RT, had no effect is confusing because the mapping of optical fiber tip locations in Figure 2D doesn't show that any fibers were placed in those nuclei. (Also, sometimes the nuclei is labeled NRT, other times RT.)

The data of Figure 2 are interpreted to mean that "These data suggest that the initiation of seizures in Ntsr1Cre;ChR2 mice requires the modulation of discrete sets of inputs that converge within the VPM rather than general, global activation of any combination of circuits within the thalamus." I would

interpret the data to show the opposite: the effective fiber placements were those located in the center and thus able to activate the largest number of VPM neurons, the ineffective experiments were when fibers were placed at the edge of VPM. Thus these data suggest that discreet circuits are LESS important than global activation of VPM.

The data of Figures 2 and 3 comparing optogenetically evoked seizures in VPM with kainic acid induced seizures are interpreted as "the similar seizure-related behaviors observed in both our Ntsr1Cre;ChR2 model and the KA model supports the involvement of VPM circuits in the manifestation of seizure-like motor abnormalities." However, the data presented clearly show that both these types of seizures generalize, producing synchronous activity across all regions of cortex. Thus, it is not possible to tell whether VPM circuits are involved with motor function because all the motor areas of the brain are active. VPM might be the seizure focus, but once the seizure generalizes the source of specific aspects of the seizure can't really be identified.

"...disorders such as dystonia are not plagued by abnormal brain activity." Abnormal brain activity is exactly what causes dystonia (and other neurological diseases that don't exhibit seizures).

"...we included any cells recorded from the thalamus and did not limit ourselves strictly to the boundaries of the VPM..." Presenting all the data collected is very reasonable and interesting, but the location of the cells recorded should be presented, Figure 4G should show cell response types based on putative location. If responsive cells are located outside of VPM it is a strong argument against the specificity of the stimulation.

The description of results shown in Figure 5, and Figure 5 itself, should be more specific than "cerebral cortex".

Discussion:

The first two paragraphs of the Discussion section are very unclear and unfocused. The findings of the paper are about the ability of cerebellar inputs to VPM to drive seizures, this should be the focus. The arguments presented making the case that optogenetic stimulation of VPM being a novel model for studying epilepsy are frankly a bit strange and not well substantiated.

The comparison between previous studies of absence seizures and the TC seizures induced here is again a bit strange. TC seizures, including those of this study, are generalized. Thus, to say that absence and TC seizures make use of overlapping networks is in a way, a given. The fact that stimulating TC seizures via VPM does NOT produce absence seizures would seem to argue that the networks are distinct.

The comparison between DBS for stopping seizures with the present protocol for inducing them is difficult to understand. It would be a bit easier if the parameters of the deep brain stimulation and the parameters used presently were listed and compared.

We would like to thank all three reviewers for providing excellent suggestions that have enabled us to strengthen our manuscript and enhance the impact of the findings. We have addressed each of the comments by altering the text, providing additional data, and revising the figures as requested.

Below are our explanations for how we have altered the manuscript in this revised version. The Reviewer's comments are written in bold, and our responses are written in regular text.

Reviewers' comments:

Reviewer #1 (Remarks to the Author):

COMMSBIO-22-2042-T

This is a manuscript that utilizes meticulously executed state-of-art techniques to determine cerebellum contributes via the thalamic VPM to trigger tonic-clonic seizures.

Unfortunately, the manuscript significantly suffers from lack of solid background, including previous publications on a similar topic as well as from overinterpretation of its own results.

Below I will point to most pressing concerns:

- 1. Line 43-45. Here the authors clearly talk about epilepsy and patients with epilepsy, but not patients with seizures. While recurrent seizures are almost sine qua non condition of epilepsy, a seizure in a patient does not mean epilepsy with long-term treatment. About 10% of population will have a seizure during lifetime. On the other hand, probability of developing epilepsy during lifetime is slightly under 4% (1:26). The authors should look into the 2010 Academy of Medicine publication on Epilepsy.**

Thank you for the clarification. We have altered our wording to reflect these differences and have ensured that we are consistent with the Academy of Medicine (lines 81-89)

- 2. Further, the authors need to look at the most recent ILAE classification of seizures and epilepsy to be able to provide pertinent and up to date definitions throughout the paper.**

We have modified our definitions to better reflect the ILAE classifications and have specified how seizures are categorized in lines 66-74.

- 3. Lines 50-52. There are many published models of seizures and also epilepsy besides absence seizures.**

Thank you for this comment. We revised this sentence to note that there are many different seizure classifications and have included a brief description of their diagnosis according to the ILAE (lines 83-86).

- 4. Lines 56-57, while the authors should be praised for their effort to link thalamic nuclei to tonic-clonic seizure generation, they need to look as papers from 80s from Browning and Nelson, who were able to elicit tonic-clonic seizures (but not pure clonic with preserved**

righting ability) after intercollicular brain stem transection, clearly indicating that all necessary circuitry for these seizures is located in the brain-stem. Indeed, there may be plethora of additional trigger points such as cortex, thalamus, basal ganglia, etc. Some structures may serve as trigger points, other as attenuation points (principle of deep brain stimulation for epilepsy).

Thank you for the direction – these studies are now referenced, and our logic was made more clear in this paragraph (lines 91-100).

- 5. If the authors are focusing on tonic clonic seizures, i.e. generalized seizures with motor convulsions (please see the current definitions as per #2), they should not be introducing absence seizures (generalized non-convulsive seizure with clear participation of thalamocortical circuitry) and setting the scene for absence seizures. The authors should also look into the book “Models of Seizure and Epilepsy”, 1st ed 2006, 2nd ed 2010 (Schwartzkroin, Moshe, Pitkanen etc) that will also provide very useful information on relatively recent developments while bridging the less recent information in.**

This is a great suggestion, thank you. We agree that absence seizures were perhaps too heavily emphasized given our intended focus on TC seizures. However, we wanted to justify our logic in targeting somatosensory areas due to its known involvement in other forms of epilepsy, namely absence seizures. To better clarify this confusion, we modified this section to deemphasize the category of absence seizures and pointed to these studies specifically as a point of reference, rather than a focal point (lines 81-89, 240-247).

- 6. VPM in humans is a major relay nucleus for somatosensory information from the face and contains third order neurons of the trigemino-thalamic tract. However, the authors mention somewhere that there is certain topography within the nucleus suggesting sub-compartmentalization of the nucleus (lines 156-161). This is very worth of investigating and discussing in detail. Indeed, reciprocity of VPM in humans and mice is another question as rodents have slightly different arrangements than primates (e.g., entopeduncular nc and others).'**

We agree that this was potentially a very interesting finding. However, due to lack of sufficient data to adequately explore this observation in the depth that it would deserve (which, ultimately, would be a separate paper), we removed its mention and will fully quantify this intriguing localization in a subsequent study.

- 7. Lines 70-71 – this is very important piece of information as in acute seizure models (PTZ, picrotoxin, bicuculline, 3-MPA etc, etc, etc)..., there is, in adult subjects, a continuous progression from “absence” seizures (freezing with electrographic spindles of epileptiform activity) through clonic seizures (face and forelimb clonus with preserved righting ability, EEG Spike and wave, polyspike and wave) eventually to tonic-clonic seizures (loss of righting ability) and this should be mentioned/emphasized. Again “Models of Seizures and Epilepsy” will provide solid background here.**

This is a great point that we have now emphasized in lines 242-247!

- 8. Lines 79-80, the reference does not seem to match cerebellar stimulation. The authors should look up the original papers of Chkhenkeli and coll**

Apologies for the incorrect reference and thank you for catching this error. This has been fixed (line 259).

- 9. Last paragraph of introduction contains unnecessary repetitions on cerebellar stimulation that should be deleted.**

We have altered the introduction to be more succinct.

- 10. Lines 136-137 do not describe the tonic-clonic seizure phenotype**

We have changed our categorization throughout the paper to indicate a generalized motor seizure rather than a tonic clonic seizure, as we recognize that the animals do not always lose their righting ability.

- 11. Lines 168-169. Primary seizures elicited by KA are NOT tonic-clonic seizures. KA induces focal seizures first with secondary generalization to clonic seizures. Only with high doses this seizure type may progress to tonic-clonic seizures (again Models of Seizures and Epilepsy will be very helpful here).**

Thank you for pointing this out. We have changed our classification to “generalized seizures” to better reflect more accurate terminology.

- 12. A clear question to the lines 163-178 is: If you lesion VPM, can you still induce tonic-clonic seizures using high dose KA or other model of tonic-clonic seizures? A simple experiment to be done, not even requiring animals with tagged neurons/circuits. Lines 335-350 – lidocaine in VPM would help!**

This is a very interesting consideration that we have also contemplated! Our intention in using the kainic acid model was purely for visual and EEG comparison of rodent generalized seizures to confirm that the phenotype we elicited was indeed reflective of a convulsive seizure, and do not suggest that KA works necessarily by the exact same pathway (via VPM).

However, we did perform the suggested experiment and found no difference between VPM-lesioned mice and sham-lesioned mice (those that underwent the surgery, but did not receive lesioning current) when delivered a systemic administration of kainic acid. Please see the figure below:

Sham vs. VPM Lesion, Racine Score over Time

Due to the length of the current paper, we chose not to include this experiment in our results, but clarified our intentions on using the KA model as a phenotypic comparator rather than a mechanistic one.

We also considered lidocaine to the VPM as an additional control, and appreciate the suggestion! Unfortunately, we could not perform this experiment, as we were unable to implant a cannula for lidocaine delivery in the same location as our fiber optic implant for stimulation. We now addressed this shortcoming in the paper for clarity (lines 798-800).

13. Lines 182-194 can be substantially shortened as EEG is a distinctive technique for positive diagnosis of epilepsy seizures.

We have shortened this paragraph to be more succinct (lines 448-453).

14. Lines 196-207, an inset in the figure with shorter time scale (to see the character of the ECoG activity) would be beneficial.

We have added an inset with a shorter time scale.

15. Wherever the authors speak about full tonic-clonic seizures in their manuscript, they need to be sure that they have observer wild run at the onset (sometimes not present), loss of posture, tonic stretch on fore and/or hindlimbs followed by a long clonus without righting present.

Although we do see progression to full TC seizures, we have altered our wording throughout the manuscript to instead classify our observations as “generalized motor” seizures.

16. Lines 387 and previous, optogenetics starts also with previously healthy brain....one of the shortcomings of chemically/electrically induced seizures. Moreover, may so called “cryptogenic” cases have genetic underpinning....

We have modified our wording to better reflect that starting with a healthy brain is a strength of the model rather than a drawback. Instead, the shortcoming of these two models is the introduction of secondary effects (lines 867-880).

17. The discussion on cerebellum should be significantly shortened.

We have removed the first paragraph on historical evidence of cerebellar contribution to seizures, and instead focused on our own results.

18. Lines 510-512 – cerebellum can very likely bypass the connections to VPM if those are severed/disable and produce tonic-clonic seizure pattern using an alternate circuitry.

This is an excellent point. We rephrased this sentence from “cerebellar input to the VPM may play a critical role....” To “the cerebellum likely plays a critical role...” (line 1196).

19. Was there any sex difference in the parameters followed?

We did not observe any differences between male and female seizure induction, and briefly added this to the methods section of our paper (lines 1253-1256).

20. Table 2 – modified Racine scale: Pinel and Rovner already in 70s modified Racine scale in a sensible way to include tonic-clonic seizures. Please find the reference (Models of Seizures and Epilepsy can help)

We added in and referenced the modified Racine scale (Lüttjohann et al, 2009) that includes TC seizures (line 1340). Thank you for this suggestion!

21. Why was Dunnett’s test used? I believe you ran Kruskal-Wallis first with post hoc Dunnett’s, correct? Rationale for statistics should be present as well as detailed description how the statistics was performed should be given. Wherever ANOVA results are presented, please show actual F values, degrees of freedom, and actual p value in a common annotation.

A Dunnett’s test was used because we aimed to compare the mean values of multiple groups to that of a control group (in the fluorescence analysis, the control was the VPM, whereas in the lidocaine experiment the control was the ‘before lidocaine’ [baseline] condition). We have now added this explanation (lines 1411-1412) and included the f values and p values for each analysis.

Reviewer #2 (Remarks to the Author):

This manuscript by Beckinghausen et al. describes an elegant study in which they designed a relatively novel mouse seizure model to demonstrate the ability of the ventral posteromedial nucleus of the thalamus (VPM) to induce a tonic-clonic (TC) like seizure. The model is based on an optogenetic approach in which they drove cell-specific expression of the excitatory opsin ChR2 in VPM afferents originating from the neocortex and cerebellum, which they beautifully demonstrate. By stimulating these fibers optogenetically they induce mice to display a profound seizure that includes several, previously described behavioral phenotypes characteristic of TC. The authors also provide data describing the heterogeneity of the VPM response during seizures using in vivo electrophysiological methods. Most strikingly, the authors demonstrate that by silencing the cerebellum using lidocaine, seizure induction is blocked. Overall, these results will be of significant interest to neurologists and neuroscientists looking to understand the mechanisms of seizure generation and systems level interconnectivity between the cerebellum-thalamus-and-neocortex. The overall design of the experiments is robust. The manuscript would be greatly enhanced by addressing several concerns related to both the description of the experiments, the interpretation of results, and the accuracy of the language used throughout the text. The two most pressing concerns are the inadequate analyses of the EEG and ephys. data and the lack of clarity in the description of the design and impact of the lidocaine experiments. My major and minor concerns are included in the pdf attachment.

Major concerns

1. **Lines 43-48:** Given that the physical expression of seizures is highly variable overall (from rigidity to rapid motor activity), at least some description of the types of motor phenotypes induced by seizure activity should be included. One suggestion "...pressing need to define the relationship between abnormal neural activity and seizure induced behavior." Also, since seizure activity is by definition not normal, it is not necessary to describe it as "abnormal" seizure activity.

Thank you for the suggestions! We have changed our wording as suggested, removed the adjective "abnormal," and included a brief descriptor of seizure classifications and phenotypes in paragraph 2 (lines 81-89).

2. **Why was 30Hz chosen for the stimulus?**

30 Hz was chosen because it was the lowest frequency that consistently elicited convulsive seizures. We have now specified this when we introduce the paradigm in paragraph 3 of the results section (lines 363-367).

3. **Lines 50-75:** The logical flow of these paragraphs should be improved to provide a more coherent rationale for investigating the hypothesis that abnormal activity in the VPM is causally related to tonic-clonic seizures. Identifying absence and TC seizures as 2 of 6 different types of generalized seizure at the outset would help; this will enable a broader audience to put into context the listed prior research as it pertains to the major classes and phenotypes of seizures. The flipping back and forth between absence and TC seizures is particularly confusing without placing into context how their similarities and differences should be considered in understanding their cellular and anatomical origins. It isn't clear from the introduction whether the prior research points to the

facial brain regions as areas especially sensitive too, or causally related to the behavioral motor defects.

We agree that the introduction and discussion of absence seizure introduced some confusion. We therefore modified our description of its relation to motor seizures by deidentifying absence seizures themselves and merely pointing to the relationship between facial regions and seizures in prior studies. Please also see our changes that we made in response to Reviewer #1. We believe the new wording of this paragraph, and the whole Introduction section in general, is much more straightforward (primarily lines 81-89).

- 4. There is an incongruence in what the overarching hypothesis and main findings are across the manuscript that should be harmonized. For instance:**
 - i. Abstract: “Here, we test the hypothesis that a facial region of the thalamus, the VPM, is a source of convulsive, tonic-clonic seizures.”**
 - ii. Introduction: “We propose that cerebellar connectivity is essential for TC seizure generation”**
 - iii. Conclusion: “we devised an optogenetics mouse model to demonstrate that 30Hz coactivation of cerebral cortical and cerebellar input to the VPM induces severe TC seizures”**

Thank you for noting this inconsistency. We have modified each of these sections to better reflect synchrony in ideas.

- 5. Line 102: You don’t actually validate the prior reports and thus the statement “We validate” should be revised. Your results show that Nstr1 is not expressed in the thalamus as previously suggested, but that cells that do express it project to this region. I suggest rewriting the section between Line 102 and 107 to more clearly orient the reader to the subsequently described results.**

Thank you for spotting this. We have adjusted these lines to be clearer and more specific about our results (lines 277-281)

- 6. Line 117: I’m confused by the first sentence as the evidence that the fluorescently labelled fibers in the thalamus originate from the cerebellar nuclei isn’t detailed until Figure 5, thus I think this statement is premature.**

We completely agree, and therefore removed this sentence.

- 7. Was there significant heterogeneity of the morphology of cerebellar nuclei neurons within each subregion?**

We did not notice any distinctive differences in neuronal morphology between nuclei and have now noted such in paragraph 2 of the results section (lines 351-353).

- 8. Line 129: The YFP fusion protein should be indicated for the strain and the use of ChR2 and ChR2-YFP should be examined for consistent use throughout.**

Thank you for suggesting this point. We have now specified the strain used in our methods section and now use Chr2-EYFP when referring to this strain throughout the paper.

9. A diagram detailing the normal progression of the seizure both during and after the stimulation would be helpful. Reference to the daily optogenetic stimulation paradigm should, at a minimum, be referred to in the text with a directed reference to the methods. The details are important and feel buried at present.

This is a great suggestion. We have now included the following figure detailing the progression of seizures and how they relate to the associated EEG. These seizures with the coinciding EEG can be seen in real time in Video 1. We have also included more references to the optogenetic paradigm.

10. The lack of statistical analyses of the coherence in Figure 3d is a weakness. The authors indicate that the program used to perform the coherence tests doesn't provide raw data in the methods...(Lines 663-665) The interpretation of this data must be carefully considered in the face of this fact. Other than a single "representative" set of graphs of coherence there is no quantitative data to backup the statement "this region [cerebellum] exhibited the greatest

increase in coherence to the ipsilateral hemisphere when comparing before-seizure to during-seizure periods.” While the graphs appear to back this up, it’s not sufficient. It’s also not clear if these graphs are an average across multiple stimulation bouts or just a single stimulation for the one animal. Would it be possible to measure the bar heights and deduce the coherence values to enable a more thorough quantitation of the results? If done carefully, and detailed in the methods, this would be a reasonable way forward for this important experiment.

Thank you for the suggestions! We utilized plot digitizer to measure the bar heights and perform a more thorough quantitation of the results and provide stronger statistical analyses (line 1421, Table 2).

11. To what extent is the EEG and motor activity correlated to the 30Hz stimulation itself. I wonder if the initial seizure rhythmicity is generated by the frequency of the stimulation and if the rebound is a rebalancing of the cerebellar and/or thalamic circuit moving back to a homeostatic state of activity that is now no longer synchronized by the optogenetic pulses. Some consideration of the underlying neural signaling activity in the Discussion section would be beneficial. The authors might consider the possibility that the seizure is caused by the impact of backpropagation of the signal rather than its stimulation of the thalamus. Is the disruptive impact local within the VPM, or back in the cerebellum (and then maybe backdown again; see comment 19).

This is a very interesting point to consider. The responses in the first few seconds of stimulation are typically much less than 30 Hz, and do not seem to have a consistent pattern. You raise an important point regarding the backpropagation, and as suggested, we provide some text in the Discussion related to this point (lines 1070-1092). Please also see our responses to your comments below regarding backpropagation as we go into a little more depth below in describing our data. In addition, please also see our responses to Reviewer # 3 as they have raised the same question.

12. Line 242-244: It isn’t clear what is meant by “seemingly tightly coupled to seizure origin.” What is being considered the seizure origin in this case? To what extent is the EEG signal in the cerebellum generated by backpropagating signals from their axons in the VPM during stimulation? If it is most, then is this finding surprising?

We apologize for the lack of clarity. We have changed this wording to “highly coherent” rather than “tightly coupled” for clarity (line 548). In the Discussion section, we also now address the possibility that backpropagating signals (initiated by the optogenetic stimulus) could also contribute to the induced behaviors – however, because the backpropagating signals seem to mostly impact cerebellar nuclei neurons themselves where they loop back out again to the same outflow tracks that were targeted in the first place and when considered together with the lidocaine experiment, our data still support the idea that cerebellar activity is important to seizures (lines 1070-1092).

13. Please clarify and potentially edit the sentence in lines 250-253. It’s not clear what the concern the authors are trying to relay through to line 256 is.

We have re-written this sentence to clarify our concern, which is that using light to directly induce neural activity is, of course, artificial. Therefore, because these neurons are being forcibly activated, the initial patterns of activity may not be entirely reflective of the initial neuronal activity that occurs naturally in seizure patients (lines 616-625).

14. It is unclear what intensity of light was used or how this was chosen. A reference to 5V was indicated in the methods, but this doesn't provide a measurement of the light's intensity for comparison to other publications.

Thank you for catching this. We now included the intensity of light both when introducing the paradigm in the results section as well in our methods section.

For example, the Methods section now reads:

However, despite the visual differences, all stimulations were kept consistent at 30Hz. Maximum LED power at the end of the implanted fiber was measured to be ~3.6 mW and stimulation consisted of this maximum brightness to induce seizures lines 1332-1334).

15. A more thorough examination of the ephys. data would enhance the manuscript. Overall, the section describing the ephys. study was confusing. It would help to initially define what the breadth of overall response properties was for the subthreshold and seizure inducing stimulations. First, it's not clear what proportion of the 70 cells was responsive to the subthreshold stimulation or what the diversity in response was (my interpretation is that Fig. 4g describes just the suprathreshold responses). If the Ntsr-ChR2 inputs are restricted to the VPM, I would expect that the recordings would support this based on their location if it was determined with sufficient resolution. Second, in Lines 279-282 it's not clear what frequency of activity was considered during these time periods. Third, it's unclear, or I missed it, whether the response properties of the subthreshold responses correlated with the suprathreshold (e.g., fired to one, but not the other). Finally, it would be interesting to know if the neural responses correlated with the underlying 30Hz frequency (I'd expect that to be the case if Ntsr-ChR2 axons were entrained to the stimulation and driving the response). Figure 4b refers to entrainment, but it is unclear if VPM neurons "entrain" to the 30Hz pulse.

Thank you for detailing this source of confusion. We have modified our ephys section to address your concerns, and hope it is now clear. We added a graph to demonstrate the number of responsive and unresponsive cells we recorded in the "subthreshold group" (n=16 neurons). These neurons were not included in the pie chart of Fig 4g, which we now explicitly state for clarity.

Ntsr1-ChR2 positive fibers are not restricted to the VPM- they innervate several nuclei of the thalamus. However, based on our experimental data, those Ntsr1-ChR2 positive fibers that converge within the VPM specifically are the ones we argue to be crucial for seizure initiation. Therefore, those neurons that were unresponsive to light stimulation were not necessarily located inside or outside the VPM – instead, we propose that these thalamic neurons are simply not innervated by ChR2-positive fibers. Noninnervated thalamic neurons are presumably located sporadically within the VPM as well as throughout the other thalamic nuclei.

For lines 279-282 (now 548-551), we analyzed the difference between the thalamic activity before the stimulation paradigm compared to the frequency after stimulation – our analyses showed no difference in frequency. Therefore, we conclude that the increases seen during stimulation were transient when applied for short durations. Thus, we did not analyze a specific frequency, but rather whether the average frequency was changed.

For comparison of subthreshold-responding cells vs. suprathreshold responding cells: unfortunately, these responses could not be correlated with one another, as many recorded neurons were lost after a few second of stimulation due to the dramatic motor convulsions that were induced (and therefore were not analyzed for suprathreshold stimulations). The neurons recorded in the subthreshold group were a separate group than those recorded for suprathreshold stimulations.

Address the possibility that backpropagation of the Ntsr-ChR2 axons after stimulation might lead to stimulation of other cerebellar projection sites. That is, do single cerebellar nuclei neurons bifurcate to multiple brain regions? Figure 5d and e seem to suggest this is the case (although I'm not clear I understand the origin of these images based on the figure legend, which suggest these are from the TdTomato mice?). If they do bifurcate from the cerebellar nuclei into the forebrain, the ramifications of backpropagation induced by optogenetic stimulation should be discussed.

These are all excellent suggestions, thank you. Reviewer #3 also had some similar comments. We have rewritten the section of the Discussion and have now provided more detail. We feel that we now provide the reader with a more balanced set of possibilities that could occur in our paradigm. The new section now reads:

Although a different paradigm than optogenetics, we previously established that low frequency deep brain stimulation (0-13Hz) to the interposed nucleus alleviates ataxia, whereas the same stimulation protocol provided in the absence of input from upstream Purkinje cells does not result in recovery⁷⁸; thus, retrograde cerebellar effects can indeed have implications on downstream circuits such as those integrated with the VPM. It is possible that in addition to the optogenetic activation of anterograde signals in the stimulated axons and terminals within the VPM, the same stimulated axons could send induced retrograde signals that travel back into the cerebellar nuclei. From there, they could loop through the cerebellar cortex and exit via the same tract, or because at least some cerebellar nuclei projections bifurcate to innervate multiple regions (Ruigrok and Teune, 2014), the induced signal could ultimately impact more than just the VPM. Moreover, en passant axons could also be activated in our stimulation paradigm. However, it is not known whether the strength of these alternate pathways would provide the same influence on the circuit and behavior as the anterograde activity that we examined by electrophysiology near the source of light in the VPM (Figure 5). In either scenario, high coherence between the cerebellum and cerebral cortex may not be completely unexpected: nonetheless, when considered in conjunction with the lidocaine blocking experiments, it is likely that cerebellar activity plays an active role with the VPM in inducing seizures (regardless of the route that takes to get there).

Please note though, current thought in the field is that indeed the cerebellum does send substantial projections to many regions that were once to contain few to no synapses from the cerebellum – for example, the VTA. However, there is no evidence of cerebellar nuclei collateralization directly into the forebrain in mammals, although some connectivity has been found in birds. Therefore, we will not dig

into this literature here as this area of cerebellar neuroanatomy is still emerging with interesting data, but certainly not enough to deliberate upon regarding functionality or disease relevance.

16. The word ablation should be replaced with a more appropriate term like silencing in relationship to the application of lidocaine.

Thank you for the suggestion. We have replaced all instances of “ablation” with “silencing”.

17. The relative timing of the application of lidocaine and the stimulation of VPM inputs and seizure induction should be more explicitly stated. It’s not clear what the duration of the 30Hz stimulation is or what time points during that stimulation the videos display (which certainly illustrate a robust result). The challenge of piecing together exactly what was done makes it a challenge to interpret the experiments; the method section unfortunately provides no additional insight.

We apologize for the confusion! We added more details to this section as well as the methods to make the experimental paradigm clearer (lines 361-369, 1313-1334).

19. The robust results of the lidocaine experiment are rather amazing. However, the authors should more clearly define how the lidocaine impacts the circuit. The main issue to be considered is whether the lidocaine has global or local effects on cerebellar signal output. Since lidocaine blocks action potentials locally by inhibiting Na channels, it is likely to leave axons in the VPM open to optogenetic stimulation, where the lidocaine is not present. The authors must consider this and factor it into their discussion of how the seizure is mechanistically prevented after silencing the cerebellum but not the neocortex.

This is a great point to consider. We have now considered and discussed both options in more detail in the revised text, whether cerebellum and its distal axons are blocked or the local cell bodies remain inhibited while their distal axons can be activated (lines 1080-1163).

20. I would encourage the authors to weave reference to their seizure model into the first paragraph of the Discussion. As it is now, there is a paragraph and a half of text before any reference to the study’s results or the impact of its findings or devised model. This could be as simple as stating from the outset, “we devised an optogenetic model to determine the role of the VPM in the generation of seizures that overcomes several shortcomings of prior models.....”

Thank you for this suggestion, this is excellent and very helpful. We have adjusted the first paragraph of our discussion to be more succinct and immediately address our own model (lines 867-880).

Minor concerns and suggestions for improvement:

- 1. Abstract: define VPM**
- 2. Line 44: consider “chronic” instead of “persistent”**
- 3. Line 50: “majority” and “primarily” are redundant in this sentence**
- 4. Line 53: this is the most common class of “generalized” seizure**
- 5. Line 72: “afferent inputs” is redundant**
- 6. Line 81: “ethical dilemmas” is vague?**

7. Line 81-83: rephrase. The variability didn't arise from the fact that the complexity of the cerebellar circuit was unknown but from the inability to consistently apply reliable stimulation.-Moreover, the facts stated are more in line with why researchers stopped investigating the cerebellum as a site for therapeutic intervention, not seizure locus.
8. Line 85: list a couple representative disorders
9. Line 94: I recommend replacing "light pulses" with "optogenetic stimulation of VPM inputs elicits...."
10. Line 97: I suggest "in the seizure network for optogenetic stimulation"
11. Line 96: replace "nuclei" with "subregions"
12. Line 96: replace "not fully understood" to "well defined"
13. Line 98-99: Consider rephrasing to highlight the brain distribution
14. Line 103: consider rephrasing to: Sun1...), which expresses a small nuclear protein in the...
15. Line 105-106: consider removing "due to the intensity of its fluorescence emission" and replace with "including distal axons located in downstream brain regions". In the next sentence, it would then be clearer to state that unlike the TdTomato reporter line, no fluorescence was observed in the thalamus of the Sun1 line. Detailing the juxtaposition provides greater context for the further examination of the Sun1 line outside the VPM. This is very clearly presented in Figure 1 a and f.
16. Figure 1 legend
 - i. Indicate the fluorescent images LUT has been inverted ii. Confirm μ is being used and not u
17. Line 121: nuclei no comma
18. Video 1 legend: "Representative video"; duration of the light pulse should be listed and the time between each section indicated
19. Figure 2: insert the p values within the graph and remove from legend
20. Figure 2 legend:
 - i. Consider changing the legend title as it only encompasses a subset of the overall figure
 - ii. In (c) change fiber optics to optic fiber; electrode track should be "optic fiber" tract
 - iii. Consider indicating what the "*" mean in the method's Quantification and Analyses section along with the statement the error bars represent SEM. These don't have to be repeated in each legend. iv. The error bars do not consistently have values in the image or the legend.
 - v. Unclear if there were cases in which mice had a mild seizure that didn't reach stage 6
 - vi. Increase the font size of the legends in 2d.
 - vii. Extra space before "6" and (e) is missing first parenthesis
 - viii. Highlight the time values by either making them bigger or putting them on top of a solid background ix. It is hard to see what the images are depicting at this magnification. Consider zooming in to better focus in on the specific element you want to highlight. Consider also including text in the images at the top of the arrows that indicates what they are pointing to.
21. Line 139: consider replacing "stimulation" as you go onto explore whether stimulation actually occurred.
22. Line 158: targeting of an "optic" fiber
23. Line 164: missing "a" between is and nondegradable
24. Line 192: replace "below" with "underneath"
25. Line 193: unclear why EEG is really any more simple than EcoG?

26. Figure 3:
- i. Consider changing the hue of the blue background color to something less bright or making it grey. It distracts from the traces.
 - ii. Put the details of the light stimulation above instead of below the blue box in (b) and include where the stimulation is occurring.
 - iii. There doesn't seem to be a reason for the text of the regions recorded to be in color. It makes them harder to read.
 - iv. Scale bar in (b) is missing
27. Figure 3 legend:
- i. Consider changing (blue box) to (blue shaded area)
 - ii. The location of stimulation is not indicated in the legend or figure
 - iii. It is unclear what "(left)" is referring to? Maybe I'm just missing something...?
28. Lines 217-219: This sentence may be better suited to the figure legend
29. Line 229: add "abnormal motor" before behavior
30. Line 246: afferent input is redundant
31. Line 249: replace "fibers" with "axons" to not confuse with optic fibers
32. Line 251: replace "direct light stimulation" with "direct stimulation of VPM inputs"
33. Line 255: delete "s" in optogenetics
34. Line 258: Ntsr1Cre;Chr2 mice
35. Line 257: remove reference to "pulling" glass electrodes, just state "using glass electrodes we recorded single unit.."
36. Line 261: replace "clean" with "stable"
37. Line 286-288: consider moving to methods
38. Figure 4:
- i. Label the red trace, it's not described in the figure or legend
 - ii. Put a title above b describing the stimulus. Do the same for c-f and consider combining these into a single panel calling them ci, cii, etc.
 - iii. Increase the text size of the descriptors on the pie chart in 4g
 - iv. Indicate that these are just for the suprathreshold
39. Figure 4 legend:
- i. Indicate what frequency is being examined in title

I recommend moving the legend here to the first two traces on the right. The reader then doesn't have to hunt for what the origin of the traces is. I would do the same for denoting the behavior as well. Maybe label it "Duration of observed behavioral seizure".

It isn't clear why the EEG signal response extends past the opto stim. in (c) but not (b). It is also surprising that the behavioral seizure outlasts the neural seizure?

Is the "ipsilateral cortex" the same as the "ipsi motor cortex"? Why name it differently.

For (d) the color scheme is hard to look at. Especially the blue on my screen. It would be helpful if the legend on the figure indicated what the two bar colors represent. Maybe decrease the width of the bar borders? Space is wasted by writing "coherence" horizontally rather than vertically (If the program forces this, then just crop it and place it vertically). The text is also very close to the panels on the left, confusing things a bit. Unclear what the dashed vertical line is indicating (not referred to in legend. I would move these graphs closer together

vertically and add a graph that quantifies the coherence differences between each comparison to highlight the differences across these comparisons.

We have changed the colors to be more intense to hopefully make it a little easier on the eye. We also moved the graphs closer and created a graph to highlight the differences between each comparison. We tried to create this graph a number of different ways, but with 50 frequencies, four brain regions, and five sets of comparisons, all of our summary graphs were far too complex to include, so we took an average of the % differences and measured the area under the curve to create a simplistic snapshot visualization. However, we chose not to include statistics on this averaged graph because we averaged averages and did not want to generate misleading statistics.

ii. “target” should be “targeted”

iii. 2 second light pulses instead of optogenetic deliveries

iv. A lighter blue consistent with the prior figure would be good v. The frequency of stimulation is not listed for 4b but is for 4c-f

40. Line 312: revise title to improve clarity

41. Line 504: I suggest rephrasing this sentence to better encompass the totality of why the model was designed. I don’t think it was designed solely to demonstrate what happens when a 30Hz light pulse is emitted on the VPM.

42. Methods:

- i. The genotype wasn’t determine by “sampling” the tail or ear. Indicate the method.**
- ii. Where did the Chr2 mice come from?**
- iii. How were the genotypes confirmed with immunohistochemistry?**
- iv. Line 553: unclear why the timed pregnancy or establishment of the embryonic day 0.5 is pertinent. v. Was sex not considered as a biological variable at all? What is the rationale for combining results from both sexes**
- vi. Line 571: “super”?**
- vii. Line 580: did you use superglue or VetBond?**
- viii. It’s unclear what the duty cycle is for the laser stimulus**

We used an LED light source, not a laser. Apologies for not being clear on our setup – we now clarified this in the methods (lines 1333-1334).

We did not find sex differences. We have included a note in the methods (lines 1253-1256).

ix. Abbreviate KA at first use and use KA from thereon

x. Was the second dose of KA the same as the first?

xi. Did the same person rate all the seizures for consistency? xii. What thickness were sections cut for immunohistochemistry xiii. Line 647: consider replacing “trained” with “habituated”

xiv. Line 650: add borosilicate

xv. Line 767: last sentence is cut off

We will not go through each of the minor points suggested, but suffice it to say that we made all the minor edits that have been addressed.

Please also note that in some cases, we did not make the recommended changes indicated above if it made more sense to completely rework the section or eliminate it all together, based on the suggestions from the other two reviewers.

Reviewer #3 (Remarks to the Author):

I have reviewed the manuscript “The cerebellum contributes to tonic-clonic seizures by altering neuronal activity in the ventral posteromedial nucleus (VPM) of the thalamus”, submitted to Communications Biology. This paper uses in vivo optogenetics and electrophysiology to examine whether a specific thalamic nuclei, VPM, and inputs to those neurons can cause tonic-clonic seizures. As the authors review, the role of the thalamocortical circuit is well-studied in absence epilepsy, but the role of the thalamus, and in particular its inputs from the cerebellum, are less understood in general and in other forms of epilepsy. The major finding that specifically silencing cerebellar inputs during seizure-inducing stimuli almost completely blocks the effect is intriguing.

Overall the study is well founded and the experiments are well executed. The general area in which this manuscript could be improved by revisions is in some of the interpretations of the data and in the organization of the discussion section. Below are my specific comments, but the manuscript would be improved by a careful review of the wording of the entire document to ensure that the statements made are consistent with the findings presented.

- 1. First is what seems to me the most important caveat to this study. Very briefly mentioned in the Discussion section is that “This analysis does not, however, exclude the possibility of antidromic activation of the cerebellum, which may partially account for its engagement.” The discussion of this topic needs to be addressed in a much more expansive way. Activation of cerebellar nuclei axons in VPM will not only activate the cerebellar nuclei through antidromic action potentials, it will also activate all the other targets that are contacted by en passant synapses or branching axons. This is true for axons from other inputs as well. Thus, even if the light stimulation is confined to VPM it may be activating every other downstream target of the brain areas that project to VPM. This does not make the findings that blocking cerebellar inputs blocks seizures any less interesting, but the authors should be wary of describing their stimulation as specific to VPM.**

Thank you for this comment, this is very important. We have added new information to the Discussion as recommended, and the language to raise these issues in more detail. The new section now reads:

Although a different paradigm than optogenetics, we previously established that low frequency deep brain stimulation (0-13Hz) to the interposed nucleus alleviates ataxia, whereas the same stimulation protocol provided in the absence of input from upstream Purkinje cells does not result in recovery⁷⁸; thus, retrograde cerebellar effects can indeed have implications on downstream circuits such as those integrated with the VPM. It is possible that in addition to the optogenetic activation of anterograde signals in the stimulated axons and terminals within the VPM, the same stimulated axons could send induced retrograde signals that travel back into the cerebellar nuclei. From there, they could loop through the cerebellar cortex and exit via the same tract, or because at least some cerebellar nuclei projections bifurcate to innervate multiple regions (Ruigrok and Teune, 2014), the induced signal could ultimately impact more than just the VPM. Moreover, en

passant axons could also be activated in our stimulation paradigm. However, it is not known whether the strength of these alternate pathways would provide the same influence on the circuit and behavior as the anterograde activity that we examined by electrophysiology near the source of light in the VPM (Figure 5). In either scenario, high coherence between the cerebellum and cerebral cortex may not be completely unexpected: nonetheless, when considered in conjunction with the lidocaine blocking experiments, it is likely that cerebellar activity plays an active role with the VPM in inducing seizures (regardless of the route that takes to get there).

Results section:

- 2. As a general comment, I think that the information found in Figure S1 is important enough to include in Figure 1. Some of the panels could be removed as the lower magnification images (Fig S1 panels D-F and J-L) do not add more information than is contained in the higher magnification images.**

Thank you, we played around with this figure a lot to try this format. In the end, we decided to remove the lower magnification images. Although we tried to move some of this information into Figure 1, it became a bit crowded, so we decided to keep S1 as a supplementary figure.

- 3. With the methods section pushed to the end it helps the reader to have some details included in the Results section. The statement “We found that unilateral delivery of 30Hz pulses of light to the ventral posteromedial nucleus (VPM) elicits a robust, TC seizure phenotype...” understates the nature of the stimuli by not including the context that the 30Hz light is delivered for an hour a day for multiple days.**

We have added the word “daily” to establish that these stimulations are repeated (line 364, 1400)

- 4. The specificity of VPM stimulation is not very convincing (even beyond the issue of activating other targets of branching axons mentioned above). ChR2 expression is very high in neighboring nuclei (presumably from axons/terminals), and unless the authors discuss the penetration of light through this tissue it is hard to imagine that those nuclei are not also activated either by light or by antidromic activation of collaterals of these axons. Additionally, the claim that stimulating the surrounding nuclei, in particular VPL and RT, had no effect is confusing because the mapping of optical fiber tip locations in Figure 2D doesn’t show that any fibers were placed in those nuclei. (Also, sometimes the nuclei is labeled NRT, other times RT.)**

We have added the targeting of adjacent nuclei to our Figure 2d to better demonstrate the specificity. We hope this clarifies that adjacent stimulations in areas with very high ChR2 expression do not elicit seizure phenotypes. We also fixed the labeling to be consistent for the reticular thalamic nucleus.

5. **The data of Figure 2 are interpreted to mean that “These data suggest that the initiation of seizures in Ntsr1Cre;Chr2 mice requires the modulation of discreet sets of inputs that converge within the VPM rather than general, global activation of any combination of circuits within the thalamus.” I would interpret the data to show the opposite: the effective fiber placements were those located in the center and thus able to activate the largest number of VPM neurons, the ineffective experiments were when fibers were placed at the edge of VPM. Thus these data suggest that discreet circuits are LESS important than global activation of VPM.**

We understand your excellent point. We have revised the language to include the ideas you have suggested and also revised our own language to better state what we meant. We have therefore fused the ideas together, which we think nicely captures the conclusion of the data (lines 399-403).

6. **The data of Figures 2 and 3 comparing optogenetically evoked seizures in VPM with kainic acid induced seizures are interpreted as “the similar seizure-related behaviors observed in both our Ntsr1Cre;Chr2 model and the KA model supports the involvement of VPM circuits in the manifestation of seizure-like motor abnormalities.” However, the data presented clearly show that both these types of seizures generalize, producing synchronous activity across all regions of cortex. Thus, it is not possible to tell whether VPM circuits are involved with motor function because all the motor areas of the brain are active. VPM might be the seizure focus, but once the seizure generalizes the source of specific aspects of the seizure can’t really be identified.**

We completely agree and apologize for the overstatement. We have removed this conclusion.

7. **“...disorders such as dystonia are not plagued by abnormal brain activity.” Abnormal brain activity is exactly what causes dystonia (and other neurological diseases that don’t exhibit seizures).**

We removed this statement.

8. **“...we included any cells recorded from the thalamus and did not limit ourselves strictly to the boundaries of the VPM...” Presenting all the data collected is very reasonable and interesting, but the location of the cells recorded should be presented, Figure 4G should show cell response types based on putative location. If responsive cells are located outside of VPM it is a strong argument against the specificity of the stimulation.**

This is a very interesting comment! We would love to go back to the data and parse apart the location of each neuron versus its activity, but unfortunately the method in which cells were recorded and saved does not allow the retrospective analysis. We tried to keep our conclusions broad, but will most definitely look at methods of reanalyzing this data in the future to delineate more specific relationships between location and activity/response.

9. **The description of results shown in Figure 5, and Figure 5 itself, should be more specific than “cerebral cortex”.**

We have added “layers 5/6 of the cerebral cortex” to be more specific (line 771). These cells were labeled both caudally and rostrally throughout the cortex and did not appear to be focused in one specific region of the cortex besides in the deep layers.

Discussion:

10. The first two paragraphs of the Discussion section are very unclear and unfocused. The findings of the paper are about the ability of cerebellar inputs to VPM to drive seizures, this should be the focus. The arguments presented making the case that optogenetic stimulation of VPM being a novel model for studying epilepsy are frankly a bit strange and not well substantiated.

We have reduced the mention of optogenetics as a beneficial model and tried to better focus the discussion (lines 867-1187).

11. The comparison between previous studies of absence seizures and the TC seizures induced here is again a bit strange. TC seizures, including those of this study, are generalized. Thus, to say that absence and TC seizures make use of overlapping networks is in a way, a given. The fact that stimulating TC seizures via VPM does NOT produce absence seizures would seem to argue that the networks are distinct.

We have removed this paragraph to avoid confusion.

12. The comparison between DBS for stopping seizures with the present protocol for inducing them is difficult to understand. It would be a bit easier if the parameters of the deep brain stimulation and the parameters used presently were listed and compared.

We added in the frequency (0-13Hz resulted in recovery). However, we want to emphasize that the mention of DBS is not to draw a parallel between the protocol or stimulation paradigms, but rather than the DBS applied downstream of Purkinje cells in a different model is affected by the upstream Purkinje cell activity (adding to the argument that retrograde effects could exist). We tried to make this clearer in the text (lines 1175-1187).

REVIEWERS' COMMENTS:

Reviewer #1 (Remarks to the Author):

This is a re-review of a previously reviewed manuscript. The authors attended to all comments and concerns of this reviewer. After reading the revised version, I have no further issues.

Reviewer #2 (Remarks to the Author):

The authors do an excellent job responding to the reviewers' extensive comments (mine in particular). I believe the manuscript is greatly improved. I no longer have any major concerns.

The authors should address minor typos throughout (e.g., extra spaces, misplaced words, missing parenthesis, etc.).

The panels for Fig. 5 need to be properly updated to match the legend (i.e., labelled with Ci, Cii, etc.).

Reviewer #3 (Remarks to the Author):

I have reviewed the revised manuscript "The cerebellum contributes to generalized seizures by altering activity in the ventral posteromedial nucleus (VPM)" by Beckinghausen et al. This work seeks to understand the role of cerebellar inputs to the thalamus in driving seizures. The authors have done a commendable job in crafting this manuscript, answering each of many comments from three very comprehensive reviews. The data analysis, visualization, interpretation, and discussion have all be improved. As they have answered each point from my original critique, and have also carefully addressed all of the very thoughtful comments of the other reviewers, I could find no other major problems to be addressed, and think this manuscript is quite ready for publication. The only two very minor issues that I would point out would be that the font on the pie chart in Figure 5g could be made bigger, and the y-axis on Figure 7d should be labeled as Normalized % Seizures.